# RELDIFF: RELATIONAL DATA GENERATIVE MODELING WITH GRAPH-BASED DIFFUSION MODELS

## ABSTRACT

Real-world databases are predominantly relational, comprising multiple interlinked tables that contain complex structural and statistical dependencies. Learning generative models on relational data has shown great promise for producing synthetic data, which can unlock access to previously underutilized information and support the training of powerful foundation models. However, existing methods often struggle to capture their complexity, typically reducing relational data to conditionally generated flat tables and imposing limiting structural assumptions. To address these limitations, we introduce RELDIFF, a novel diffusion generative model that synthesizes relational databases by explicitly modeling their foreign key graph structure. RELDIFF combines a joint graph-conditioned diffusion process across all tables for attribute synthesis and a $D2K$+SBM graph generator based on the stochastic block model for structure generation. The decomposition of graph structure and relational attributes ensures both high fidelity and referential integrity, both of which are crucial aspects of synthetic relational database generation. RELDIFF achieves state-of-the-art performance in generating synthetic relational databases on 11 benchmark datasets.

## 1 INTRODUCTION

Relational databases, which organize information into multiple interconnected tables governed by foreign key references, underpin over 70% of global data management systems (DB-Engines, 2024) and form the foundation for much of today's digital infrastructure. However, unlocking access to high-quality real-world datasets is often limited by fairness and privacy concerns (Ntoutsi et al., 2020; Hernandez et al., 2022; van Breugel & van der Schaar, 2023), particularly in sensitive domains like healthcare (Appenzeller et al., 2022; Gonzales et al., 2023) and finance (Assefa et al., 2020; Potluru et al., 2024). Thus, synthetic data generation emerges as a promising solution, offering a way to preserve crucial statistical properties while effectively mitigating privacy risks (Raghunathan, 2021). Moreover, synthetic data can unlock access to valuable enterprise and healthcare databases, facilitating the creation of powerful relational and tabular foundation models (van Breugel & van der Schaar, 2024) and has shown promise in missing value imputation (You et al., 2020; Zhang et al., 2025) and data augmentation (Fonseca & Bacao, 2023).

Unlike image data, which comprises pure continuous pixel values with local spatial correlations, or text data, which comprises tokens that share the same vocabulary, tabular data includes heterogeneous and often imbalanced distributions (Xu et al., 2019), making it challenging to learn joint probabilities across multiple columns. Moreover, relational databases exhibit complex structural hierarchies and statistical dependencies across their interconnected tables, often stored at varying levels of normalization (Codd, 1970; Delobel, 1978), which further intensifies these inherent difficulties.

A common simplification involves flattening relational schemas into single tables (Ge et al., 2021; Ghazi et al., 2023), but this approach quickly becomes impractical for large-scale and complex schemas characteristic of real-world databases (Pang et al., 2024). More recent methods (Cai et al., 2023; Xu et al., 2023; Pang et al., 2024) instead model relational databases as a sequence of conditionally generated tables. This design requires a pre-specified ordering of tables and typically relies on strong independence assumptions, which constrain modeling flexibility. As a result, current methods have struggled with two key limitations: (1) the inability to generate arbitrary relational schemas and (2) the failure to effectively preserve inter-table correlations and long-range dependencies between attributes linked by foreign key relationships (Hudovernik et al., 2024).

In this paper, we propose RELDIFF, a principled generative framework for relational databases. RELDIFF enables the synthesis of arbitrary relational schemas by explicitly modeling the underlying database structure with graphs. Instead of simply decomposing the relational into conditionally independent tables, RELDIFF first utilizes a specifically designed $D2K$+SBM graph generator to preserve the cardinality of foreign key relationships and the hierarchical dependencies inherent in the relational structure. Built upon this faithful structural representation, we further define a joint graph-based diffusion model for attribute synthesis across interconnected tables, built upon a unified diffusion process modeling both continuous and discrete features. The resulting model can explicitly capture both inter- and intra-table dependencies powered by graph neural networks (GNNs), and can model the heterogeneous numerical and categorical features within each table.

The unique advantages of RELDIFF come from the key innovations in our approach, including: (1) A principled formulation for generating foreign key structures in relational databases, incorporating hard constraints to ensure *referential integrity* through a novel application of Bayesian stochastic blockmodels; (2) A joint diffusion model for synthesizing *mixed-type* attributes, conditioned on graph structure using GNNs, to better capture global dependencies; (3) We define our diffusion model in data space and explicitly consider dimension tables, a fundamental component of real-world databases, as a distinct data type. Furthermore, all the developments are general and data agnostic, without any predefined assumption of the relational data structures. These innovations allow us to model relational databases with arbitrarily complex schemas and preserve both statistical and structural dependencies of the data.

We conduct comprehensive experiments to justify the effectiveness of our proposed method. Empirical results across two benchmarks, covering 11 datasets and 8 metrics, demonstrate that RELDIFF can consistently outperform previous methods, with up to $80\%$ improvement over the state-of-the-art in preserving column correlation between connected tables. The significant improvements highlight the superior generative capacity of our approach on relational data.

## 2 RELATED WORK

**Relational Database Synthesis.** Patki et al. (2016) were the first to propose a learning-based method for relational database synthesis - the Synthetic Data Vault (SDV). Recent methods broadly fall into neural network-based (prioritizing fidelity) and marginal-based (focusing on differential privacy) approaches. While marginal-based methods are established for single-table synthesis (Zhang et al., 2017; McKenna et al., 2022) , their extension to relational data is more recent, with PrivLava (Cai et al., 2023) and several newer methods emerging (Alimohammadi et al., 2025; Kapenekakis et al., 2024; Cai et al., 2025).

We focus on neural network-based relational database synthesis, preserving fidelity and utility for arbitrary schemas. Graph variational autoencoder-based methods (Mami et al., 2022) have been investigated but encounter scalability issues with real-world databases. Generative adversarial network (GAN)-based methods, such as RCTGAN (Gueye et al., 2023) and IRG (Li et al., 2024a), extend the successful CTGAN (Xu et al., 2019) architecture from single-table synthesis to relational data. However, recent advancements have demonstrated the superior performance of diffusion models over GANs in various generative tasks. Autoregressive approaches, leveraging language models (Solatorio & Dupriez, 2023) and any-order networks (Tiwald et al., 2025), have also been investigated, but their autoregressive nature makes them better suited for simpler relational structures, particularly those with single-parent schemas. In contrast, the work of Xu et al. (2023) proposes a method for generating many-to-many datasets using bipartite $2K$ random graphs. Similar to our approach, they decouple the generation of database structure and attribute synthesis. However, their graph generation method does not capture hierarchical relational structures and primarily focuses on many-to-many relationships, generating tables sequentially. Building on the success of diffusion models for single-table data generation (Kotelnikov et al., 2023; Zhang et al., 2024; Shi et al., 2025a), two diffusion-based methods for relational data have emerged: ClavaDDPM (Pang et al., 2024) and RGCLD (Hudovernik, 2024). Both of these approaches largely treat relational database synthesis as a series of conditional single-table generation tasks, relying on a pre-specified table ordering and introducing limiting assumptions about the relational dependencies. A detailed overview of related work is provided in Appendix A.

**Graph Structure Generation.** Realistic and efficient graph generation models originate from the degree sequence problem. $dK$-random graphs (Mahadevan et al., 2006) preserve the degrees of nodes in $d$-sized connected subgraphs. $0K$ graphs preserve: the density (by convention), $1K$ graphs: the degree distribution, $2K$ graphs: the joint degree distribution of neighbors, $3K$ graphs: the degrees of connected triplets of nodes. While there exist efficient methods to generate up to $2K$-graphs, generating $3K$ graphs is NP-hard. Therefore, the $2K+$ graph construction framework (Tillman et al., 2019) proposes heuristic approaches for targeting additional properties such as connected components and clustering. We extend the $2K+$ framework by preserving (not only targeting) hierarchical and any other modular organization present in relational databases. We employ the Stochastic Block Model (SBM) (Holland et al., 1983) where nodes are partitioned into blocks which define their connectivity (blocks are subsets of relational tables). The degree-corrected SBM (Dasgupta et al., 2004) accounts for the variance in node degrees, while the microcanonical version enforces hard constraints on the edges (Peixoto, 2017). A hierarchy of nested SBMs reduces the minimum detectable size of blocks from $\mathcal{O}(\sqrt{n})$ down to $\mathcal{O}(\log n)$, where $n$ is the number of nodes (Peixoto, 2014).

Deep generative models for graphs, such as diffusion (Liu et al., 2019; Vignac et al., 2023) and autoregressive models (You et al., 2018; Liao et al., 2019), exist but often assume dense representation, scaling poorly to large relational database graphs (Jang et al., 2024; Li et al., 2024b). Unlike our model, they typically do not enforce hard structural constraints relevant to our work.

## 3 PRELIMINARIES

**Notation.** We begin by introducing a formal definition of relational databases, following the RDL framework (Fey et al., 2024), which provides a principled abstraction of relational data as heterogeneous graphs. This formulation enables us to decouple the generative process into two components: (1) the generation of database structure via a schema-consistent relational entity graph, and (2) the joint synthesis of entity-level attributes conditioned on this structure and local neighborhoods.

A relational database $(\mathcal{T}, \mathcal{L})$ consists of a collection of tables $\mathcal{T} = \{T_1, \ldots, T_n\}$, and links between tables $\mathcal{L} \subseteq \mathcal{T} \times \mathcal{T}$. A link $L = (T_{\text{fkey}}, T_{\text{pkey}})$ exists if a foreign key column in $T_{\text{fkey}}$ references a primary key column in $T_{\text{pkey}}$. Each table is a set $T = \{v_1, \ldots, v_{n_T}\}$, whose elements $v_i \in T$ are called *rows* or *entities* . Each entity $v \in T$ is defined as a tuple: $v = (p_v, K_v, x_v)$ where: $p_v$ is the **primary key**, uniquely identifying the entity $v$, $K_v \subseteq \{p_{v'} : v' \in T' \text{ and } (T, T') \in \mathcal{L}\}$ is the set of **foreign keys**, establishing links from $v \in T$ to entities $v' \in T'$, where $p_{v'}$ is the primary key of $v'$ in table $T'$ and $x_v$ contains the **attributes**, representing the informational content of the entity.

The two central objects of RDL as defined by Fey et al. (2024) are the *schema graph* and *relational entity graph*. The schema graph defines the table-level structure of data. Given a relational database $(\mathcal{T}, \mathcal{L})$ and the inverse set of links as $\mathcal{L}^{-1} = \{(T_{\text{pkey}}, T_{\text{fkey}}) \mid (T_{\text{fkey}}, T_{\text{pkey}}) \in \mathcal{L}\}$, the schema graph is the graph $(\mathcal{T}, \mathcal{R})$ with node set $\mathcal{T}$ and edge set $\mathcal{R} = \mathcal{L} \cup \mathcal{L}^{-1}$. The nodes of the schema graph serve as type definitions for the heterogeneous relational entity graph.

The relational entity graph is a heterogeneous graph $G = (\mathcal{V}, \mathcal{E}, \phi, \psi)$, with node set $\mathcal{V}$ and edge set $\mathcal{E} \subseteq \mathcal{V} \times \mathcal{V}$ and type mapping functions $\phi : \mathcal{V} \to \mathcal{T}$ and $\psi : \mathcal{E} \to \mathcal{R}$, where each node $v \in \mathcal{V}$ belongs to a node type $\phi(v) \in \mathcal{T}$ and each edge $e \in \mathcal{E}$ belongs to an edge type $\psi(e) \in \mathcal{R}$. Specifically, the sets $\mathcal{T}$ and $\mathcal{R}$ from the schema graph define the node and edge types of our relational entity graph.

Real-world relational databases contain diverse data types. Following prior work, we focus on numerical, categorical, and datetime attributes, representing each as either a continuous or discrete random variable for unified modeling. We handle dimension tables explicitly as fixed-size vocabulary lookups to ensure schema consistency and improve sample quality.

**Gaussian Diffusion.** For a numerical attribute $z$, the forward diffusion process gradually perturbs the data with increasing Gaussian noise: $q_{\text{num}}(z_t \mid z_0) = \mathcal{N}(z_0, (\sigma^{\text{num}}(t))^2)$, where $\sigma^{\text{num}}(t) : [0, 1] \to \mathbb{R}^+$ is an increasing function that governs the cumulative noise level over time. The marginal distribution $p(z_0)$ at time t = 0 corresponds to the data distribution, while $p(z_1)$ converges to the know noise distribution $\mathcal{N}(0, (\sigma^{\text{num}}(1))^2)$ from which we can easily sample. Following the framework of Karras et al. (2022), the true reverse distribution $q_{\text{num}}(z_s \mid z_t)$ can be formulated by the solutions to the ordinary differential equation (ODE) $dz^{\text{num}} = -[\frac{d}{dt}\sigma^{\text{num}}(t)]\sigma^{\text{num}}(t)\nabla_z \log p_t(z)dt$, where $0 \le s < t \le 1$. To learn the generative model, we approximate the true score function

Figure 1: **RELDIFF framework overview**. RELDIFF applies forward diffusion to mixed-type attributes within each relational table and performs joint reverse denoising across tables, conditioned on the relational entity graph and node neighborhoods. Learnable embeddings handle dimension tables (e.g., *Products*), and a sampled synthetic entity graph guides the generation process.

$\nabla_z \log p_t(z^{\mathrm{num}})$ using a neural network $\mu_\theta^{\mathrm{num}}$, which is trained via the following denoising loss:

$$\mathcal{L}_{\mathrm{num}}(\theta) = \mathbb{E}_{z_0 \sim p(z_0)} \mathbb{E}_{t \sim U[0,1]} \mathbb{E}_{\epsilon \sim \mathcal{N}(\mathbf{0},\mathbf{I})} \left\| \mu_\theta^{\mathrm{num}}(z_t, t) - \epsilon \right\|_2^2 . \tag{1}$$

**Masked Diffusion.** For a categorical attribute $c$ with $K$ categories, we introduce an additional $(K+1)^{\mathrm{th}}$ to represent the special [MASK] state and denote it as $\mathbf{m} = (0, \ldots, 1) \in \{0,1\}^K$ using the one-hot representation. Let $\mathrm{cat}(\cdot; \boldsymbol{\pi})$ denote the categorical distribution over $K$ classes, parameterized by the probability vector $\boldsymbol{\pi} \in \Delta^K$. The forward diffusion process operates in continuous-time by gradually masking the original values: $q_{\mathrm{cat}}(c_t|c_0) = \mathrm{cat}(c_t; \alpha_t c_0 + (1 - \alpha_t)\mathbf{m})$, where $\alpha_t = \exp(-\sigma^{\mathrm{cat}}(t))$ is a decreasing function between 0 and 1 controlling the masking rate. Following Sahoo et al. (2024) and Shi et al. (2024), the true reverse transition distribution $q(c_s|c_t)$ is given as:

$$q_{\mathrm{cat}}(c_s|c_t) = \begin{cases} \mathrm{cat}(c_s; c_t) & c_t \neq \mathbf{m}, \\ \mathrm{cat}\left(c_s; \frac{(1-\alpha_s)\mathbf{m} + (\alpha_s - \alpha_t)q_{\mathrm{cat}}(c_0|c_t)}{1 - \alpha_t}\right) & c_t = \mathbf{m}. \end{cases} \tag{2}$$

To model this generative process, we train a neural network $\mu_\theta^{\mathrm{cat}}$ to predict the original category $c_0$ from the masked input $c_t$, i.e. to estimate $q_{\mathrm{cat}}(c_0|c_t)$. The model is optimized using a cross-entropy loss under the continuous-time limit:

$$\mathcal{L}_{\mathrm{cat}}(\theta) = \mathbb{E}_q \int_{t=0}^{t=1} \frac{\alpha_t'}{1 - \alpha_t} \mathbb{1}_{\{c_t = \mathbf{m}\}} \log \langle \mu_\theta^{\mathrm{cat}}(c_t, t), c_0^{\mathrm{cat}} \rangle dt. \tag{3}$$

## 4 METHOD

This section details **RELDIFF**, our framework for generating synthetic relational databases. We provide a high-level overview of our generative model in Section 4.1. First, we describe how we model the relational graph structure using Bayesian stochastic blockmodels, ensuring referential integrity and preserving relationship cardinalities and hierarchical dependencies (Section 4.2). Next, we present our joint relational diffusion model for synthesizing attributes across the database schema, parameterized by a single heterogeneous graph neural network with tabular transformer-based encoders and decoders (Section 4.3). Finally, we detail the training and sampling procedures (Section 4.4).

### 4.1 OVERVIEW

In line with Probabilistic Relational Models (PRMs) (Friedman et al., 1999; Getoor et al., 2001), we decompose the generative process into modeling the relational graph structure and the attributes of the tables. We treat the relational entity graph $G = (\mathcal{V}, \mathcal{E})$ as a single sample from some unknown joint distribution $p(\mathcal{V}, \mathcal{E})$. Our objective is to sample from this distribution, ensuring adherence to referential integrity constraints and the preservation of statistical dependencies. We formalize this joint sampling through the following conditional decomposition $p(\mathcal{V}, \mathcal{E}) = p(\mathcal{E})p(\mathcal{V} \mid \mathcal{E})$. This decomposition enables efficient modeling even for complex schemas. We provide a disscusion of alternative formulations in Appendix F. See also Xu et al. (2023) for a detailed discussion for many-to-many schemas.

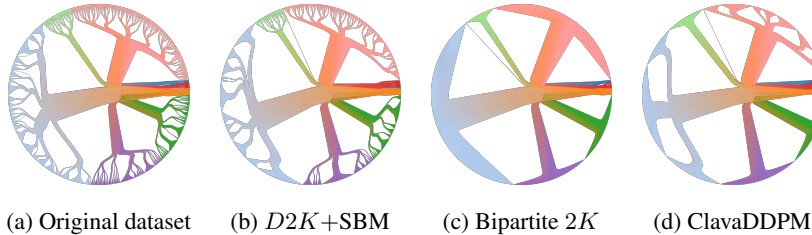

(a) Original dataset     (b) $D2K$+SBM     (c) Bipartite $2K$     (d) ClavaDDPM

Figure 2: **Hierarchical structure** of the F1 dataset. Our SBM-based method preserves the F1 dataset's foreign key graph's joint degree distribution and hierarchy. In contrast, the bipartite $2K$-graph approach (Xu et al., 2023) loses this structure despite matching degree distributions, and ClavaDDPM, by implicitly modeling the structure, retains some hierarchy but not the degrees.

To model graph structure $p(\mathcal{E})$, we present a novel approach based on established models from graph theory. Within our joint diffusion framework, we formulate $p(\mathcal{V} \mid \mathcal{E})$ to explicitly model the dependencies between nodes representing entities across different tables during generation.

## 4.2 GRAPH STRUCTURE GENERATION

To generate realistic synthetic relational structures, we focus on sampling graphs that preserve the original database size, enforce referential integrity, and respect exact table and relationship cardinalities. Modern deep generative approaches are not applicable here due to their scalability limitations with large, sparse graphs (Jang et al., 2024). Instead, we base our approach on classical random graph models, which provide principled, efficient mechanisms for sampling large, structured graphs while preserving referential integrity by design.

The graph structure of a relational database is a heterogeneous entity graph $G = (\mathcal{V}, \mathcal{E}, \phi, \psi)$ defined above. Our objective is to learn a distribution $p(\mathcal{E})$ over such graphs, conditioned on a fixed set of nodes $v \in \mathcal{V}$, the number of edges $m_r = |\mathcal{E}_r|$ of each type $r \in \mathcal{R}$ and the node degree sequence $k_v^r = |v \in \mathcal{E}_r|$, where $\sum_{v \in V} k_v^r = 2m_r$. This corresponds to fixed row counts, and exact entity and relationship cardinalities. We can opt to preprocess the entity graph by converting nodes with two parents and no children into many-to-many edges to improve computational efficiency (see Appendix E.1). This one-to-one transformation is reverted after sampling the new edge set.

To generate samples from $p(\mathcal{E})$, we utilize nonparametric Bayesian SBMs (Peixoto, 2019) as a model of graphs with the above structural constraints. The microcanonical degree-corrected SBM (Peixoto, 2017) defines the distribution $p(\mathcal{E}|b, m, k)$, where $b : \mathcal{V} \to \mathbb{Z}$ is a partition of nodes into some latent (disjoint) blocks. By setting $b = \phi$, and generating the subgraph corresponding to each foreign-key relationship independently, we obtain a $D2K$ graph generator (Tillman et al., 2017) of individual relationships. $2K$-random graphs preserve the node degree sequence and the degree correlations of neighbouring nodes (Mahadevan et al., 2006). To preserve also a global hierarchical and other modular organization of the database, we use the maximum likelihood partition $b^*$ that minimizes the description length of a nested hierarchy of SBMs (Peixoto, 2014). Note that we constrain the partition $b^*$ by node types $\phi$ such that $b^*(v) = b^*(u)$ implies $\phi(v) = \phi(u)$ for all $v, u \in \mathcal{V}$. We refer to this model as $D2K$+SBM graphs, consistent with the literature (Tillman et al., 2017; 2019).

The generation process proceeds in three stages.

1. We employ an efficient Markov Chain Monte Carlo approach (Peixoto, 2014) to infer the most likely partition $b^*$ of the edge set $\mathcal{E}$.

2. For each relationship $r \in \mathcal{R}$, we sample a new edge set $\mathcal{S}_r$ independently from $p(\mathcal{E}_r|b_r^*, m_r, k^r)$, where $b_r^*$ is the partition of nodes induced by $\mathcal{E}_r$. When $\mathcal{E}_r$ induces a simple graph, we ensure to sample $\mathcal{S}_r$ only from simple graphs (unless stated otherwise).

3. The final generated graph is induced by the edge set $\cup_{r \in \mathcal{R}} \mathcal{S}_r$.

Our approach preserves the joint degree distribution of individual relationships by construction and, as illustrated in Figure 2, retains the hierarchical structure of the relational entity graph.

## 4.3 JOINT MULTI-RELATIONAL DIFFUSION

To model the distribution over node attributes, we define a hybrid diffusion process that applies forward noise independently across entries and independently across attribute types (numerical and categorical) within each entry. Let $\mathcal{V}_t$ denote the set of all entities at time $t$, and $x_t^v$ the attribute of a single entry $v$. This forward process is given by:

$$q(\mathcal{V}_t \mid \mathcal{V}_0) = \prod_{v_t \in \mathcal{V}_t} q_{\text{num}}^{\phi(v)}(x_t^{\text{num}} \mid x_0^{\text{num}}) \cdot q_{\text{cat}}^{\phi(v)}(x_t^{\text{cat}} \mid x_0^{\text{cat}}), x_t \in v_t. \tag{4}$$

The true reverse process is then modeled as the joint posterior:

$$q(\mathcal{V}_s \mid G_t) = \prod_{v_s \in \mathcal{V}_s} q_{\text{num}}^{\phi(v_s)}(x_s^{\text{num}} \mid x_t^{\text{num}}, G_t) \cdot q_{\text{cat}}^{\phi(v_s)}(x_s^{\text{cat}} \mid x_t^{\text{cat}}, G_t), x_s \in v_s, \tag{5}$$

which factorizes over entities but allows each denoising step to condition on the full database context $G_t$ at a given timestep $t$. Note that to simplify notation, we omit defining a common space on which $q$ is defined, but make a distinction between $q^{\phi(v)}$, which are defined on subsets of the joint space.

We formulate the learning objective as a graph-based denoising task, where the goal is to train a model $p_\theta(x_s^v \mid x_t^v, G_t)$ that reconstruct clean attributes from noisy inputs. However, conditioning on the full graph at every step can be computationally prohibitive, as real-world databases typically contain millions of entries.

To improve efficiency while still capturing the interactions between attributes across the connected tables, we assume conditional independence of each node $v \in \mathcal{V}$ given its noisy $k$-hop neighborhood at timestep $t$, denoted $\mathcal{N}_k(v)_t$. Under this assumption, the model is approximated as $p_\theta(x_s^v \mid x_t^v, \mathcal{N}_k(v)_t)$ and parameterized using a GNN that operates locally over the neighborhood $\mathcal{N}_k(v)_t$.

By plugging in the objective functions corresponding to the masked and Gaussian diffusion processes (defined by eqs. (1) and (3)), we end up with the following optimization objective with two weight terms $\lambda_{\text{num}}$ and $\lambda_{\text{cat}}$:

$$\mathcal{L}_{\text{RELDIFF}}(\theta) = \sum_{T_i \in \mathcal{T}} (\lambda_{\text{num}} \mathcal{L}_{\text{num}}(\theta) + \lambda_{\text{cat}} \mathcal{L}_{\text{cat}}(\theta)) = \mathbb{E}_{t \sim U(0,1)} \sum_{T_i \in \mathcal{T}} \mathbb{E}_{x_t^v \sim q^{T_i}(x_t^v, x_0^v)}$$

$$\left( \lambda_{\text{num}} \| \mu_\theta^{\text{num}}(x_t^v, t, \mathcal{N}_k(v))^{T_i} - \epsilon_{\text{num}} \|_2^2 + \sum_{c_t \in x_t^{v,\text{cat}}} \frac{\lambda_{\text{cat}} \alpha_t'}{1 - \alpha_t} \mathbb{1}_{\{c_t = \mathbf{m}\}} \langle \mu_\theta^{\text{cat}}(x_t^v, t, \mathcal{N}_k(v))^{T_i}, c_0^{\text{cat}} \rangle \right) \tag{6}$$

We parametrize our model with a heterogeneous graph neural network with transformer encoders and decoders and one MLP backbone per table. We use a heterogeneous variant of the GraphSAGE network (Hamilton et al., 2017).

## 4.4 TRAINING AND INFERENCE

With the forward process defined in Eq. (4) we present the training procedure for our joint diffusion model in Algorithm 1. We begin each training step by sampling a subgraph that maintains the proportional representation of nodes from each table in the original database. Subsequently, we sample a timestep $t \sim U(0,1)$ using a low-discrepancy sampler similar to the one proposed by Kingma et al. (2021) and apply the corresponding noise schedule s to perturb the numerical and categorical attributes. The noisy subgraph, along with $t$, is then fed into our model. The model jointly denoises the attributes across all tables and we perform a gradient step on the combined loss function defined in Eq. (6).

---

**Algorithm 1** Training

1: **repeat**
2:     Sample batch $G_{\text{batch}}$
3:     Sample $t \sim U(0,1)$
4:     **for** $T_i \in \mathcal{T}$ **do**
5:         $x_0 \leftarrow G_{\text{batch}}.x_0^{T_i}$
6:         Sample $\epsilon_{\text{num}} \sim \mathcal{N}(0, I_{M_{T_i^{\text{num}}}})$
7:         $x_t^{\text{num}} = x_0^{\text{num}} + \sigma_{\text{num}}(t) \cdot \epsilon_{\text{num}}$
8:         Sample $x_t^{\text{cat}} \sim q(x_t | x_0)$
9:         $G_{\text{batch}}.x_t^{T_i} \leftarrow [x_t^{\text{num}}, x_t^{\text{cat}}]$
10:     **end for**
11:     Take gradient descent step on $\nabla_\theta \mathcal{L}_{\text{RELDIFF}}$
12: **until** converged

---

Relational databases often consist of millions of entities, making it infeasible to load and process the full relational graph in memory during training. Training on the entire graph would be ideal from a

computational perspective, since it provides complete access to the computational graph and enables loss computation across all nodes. However, it would constrain us to sampling a single noise level per training iteration, as the entire graph must be denoised simultaneously. In contrast, minibatch training provides a practical trade-off: it reduces memory consumption and enables sampling across different noise levels, at the cost of reusing only a portion of the computation graph per step.

**Subgraph Sampling for Efficient Training**. For databases organized into multiple disjoint subgraphs (e.g., those following a snowflake schema), minibatch construction is straightforward. We sample $n$ subgraphs independently at each training step and compute the loss across all nodes within them. In more general settings, where foreign key relationships form a connected network rather than isolated components, we adapt the neighbor sampling procedure of Hamilton et al. (2017). Specifically, we begin by selecting a set of seed nodes for each table, then sample their $k$-hop neighborhoods. The resulting subgraphs are merged to form a single, connected minibatch subgraph used for training.

**Sampling** During the sampling process, we first generate a new relational entity graph using our $D2K+$SBM graph generator, capturing the structural properties of the original database. To initiate the reverse diffusion, we sample initial values for numerical attributes from a standard Gaussian prior distribution and set all categorical attributes to the masked state. At each subsequent denoising step of the reverse diffusion, we jointly denoise the attributes of all tables using our learned diffusion model, allowing for simultaneous refinement and capturing inter-table dependencies. For the reverse diffusion process, we utilize a stochastic sampler, similar to the one proposed by Shi et al. (2025a), to introduce stochasticity and diversity into the generated samples.

After completing the denoising process, we transform the generated graph entities back into a tabular format, reconstructing the table structure. Finally, we add foreign keys to the tables based on the edges of the generated relational entity graph, ensuring the generated data adheres to the structural relationships defined by the schema.

## 5 EXPERIMENTS

We evaluate RELDIFF by comparing it against 6 generative methods using two relational database generation benchmarks consisting of 11 real-world datasets totaling 67 tables and 64 foreign key relationships. We focus on multi-table fidelity and downstream task performance. Additional results on single-table fidelity, privacy, and ablation studies are provided in Appendix D. In Appendix E, we analyze the efficiency of our method by reporting training and sampling runtimes and by scaling to two large relational databases—an order of magnitude larger in the number of rows and tables than those used in prior work—demonstrating RELDIFF 's superior scalability.

### 5.1 EXPERIMENTAL SETUP

**Real-world datasets.** We conduct our evaluation on eleven real-world relational databases from two benchmarks from related work. The datasets include: *Biodegradability*, *Berka*, a relational version of the *Cora* dataset, *Walmart Recruiting - Store Sales Forecasting*, *Airbnb Bookings*, *Rossmann Store Sales*, *CCS*, *Instacart 05*, and *F1*. These datasets vary in the number of tables, the maximum depth, the number of foreign-key relationships, and structural complexity. Details can be found in Appendix B.1.

**Baselines.** We compare our method with state-of-the-art methods from each benchmark. These include ClavaDDPM (Pang et al., 2024), RCTGAN (Gueye et al., 2023), RealTabFormer (Solatorio & Dupriez, 2023), SDV (Patki et al., 2016) and TabularARGN (Tiwald et al., 2025) on the SyntheRela benchmark and ClavaDDPM, SDV and PrivLava (Cai et al., 2023) on the ClavaDDPM benchmark.

**Evaluation metrics.** We follow the protocols of Jurkovič et al. (2025) and Pang et al. (2024) and use the same evaluation metrics: 1) Fidelity: Shape, Trend, C2ST, C2ST-Agg, k-hop correlation and cardinality similarity assess how well the synthetic data can faithfully recover the ground-truth data distribution; 2) Downstream tasks: machine learning efficiency using RDL utility evaluates the models' potential to power downstream tasks; 3) Privacy: The distance to closest record (DCR) score evaluates the level of privacy protection by measuring how close the synthetic data is to the training data and membership inference attacks (MIA) evaluate how much of the training data can be detected by an attacker given a sample of held-out and synthetic data. We provide detailed descriptions of all metrics in Appendix B.2.

Table 1: **Multi-table results** on the SyntheRela benchmark. For each dataset we report the average detection accuracy for C2ST-Agg (lower is better) and k-hop correlation similarity (higher is better). The number of k-hop results is determined by maximum depth of the dataset. We report the best result in **bold**. DNC denotes *Did Not Converge*. (Baselines from Jurkovič et al. (2025).)

| | Metric | TabARGN | ClavaDDPM | RCTGAN | REALTABF. | SDV | RelDiff | Improv. |
|---|---|---|---|---|---|---|---|---|
| Airbnb | C2ST-Agg ($\downarrow$) | $63.47_{\pm0.88}$ | $\approx 100.0$ | $98.22_{\pm0.08}$ | $99.13_{\pm0.02}$ | $99.94_{\pm0.01}$ | $\mathbf{55.68}_{\pm0.66}$ | 12.26 |
| | Cardinality ($\uparrow$) | $98.59_{\pm0.32}$ | $99.65_{\pm0.06}$ | $95.45_{\pm0.62}$ | $76.38_{\pm0.47}$ | $26.36_{\pm0.03}$ | $\mathbf{100.0}$ | 0.35 |
| | 1-HOP ($\uparrow$) | $79.66_{\pm0.36}$ | $86.69_{\pm0.14}$ | $68.78_{\pm0.54}$ | $33.99_{\pm5.76}$ | $24.58_{\pm0.03}$ | $\mathbf{89.37}_{\pm0.38}$ | 3.10 |
| Rossmann | C2ST-Agg ($\downarrow$) | $60.43_{\pm0.63}$ | $85.77_{\pm0.07}$ | $86.11_{\pm1.01}$ | $85.90_{\pm1.33}$ | $98.37_{\pm0.23}$ | $\mathbf{51.06}_{\pm1.39}$ | 15.51 |
| | Cardinality ($\uparrow$) | $94.17_{\pm1.84}$ | $99.19_{\pm0.29}$ | $82.69_{\pm1.95}$ | $41.82_{\pm10.29}$ | $99.16_{\pm0.15}$ | $\mathbf{100.0}$ | 0.81 |
| | 1-HOP ($\uparrow$) | $92.95_{\pm0.78}$ | $82.81_{\pm0.47}$ | $87.02_{\pm0.17}$ | $80.25_{\pm0.84}$ | $73.84_{\pm0.34}$ | $\mathbf{96.73}_{\pm0.18}$ | 4.06 |
| Walmart | C2ST-Agg ($\downarrow$) | $94.81_{\pm1.68}$ | $73.33_{\pm2.92}$ | $94.81_{\pm1.68}$ | $90.0_{\pm0.91}$ | $88.52_{\pm1.60}$ | $\mathbf{66.30}_{\pm1.68}$ | 9.60 |
| | Cardinality ($\uparrow$) | $65.93_{\pm1.98}$ | $93.33_{\pm2.28}$ | $88.15_{\pm1.51}$ | $85.56_{\pm4.57}$ | $86.30_{\pm1.09}$ | $\mathbf{100.0}$ | 7.14 |
| | 1-HOP ($\uparrow$) | $75.40_{\pm1.49}$ | $86.40_{\pm1.73}$ | $79.02_{\pm0.15}$ | $74.99_{\pm0.20}$ | $76.64_{\pm1.07}$ | $\mathbf{91.87}_{\pm0.42}$ | 6.34 |
| Berka | C2ST-Agg ($\downarrow$) | $80.56_{\pm1.86}$ | $69.12_{\pm0.63}$ | $76.86_{\pm2.22}$ | | $77.43_{\pm0.14}$ | $\mathbf{55.69}_{\pm0.82}$ | 19.43 |
| | Cardinality ($\uparrow$) | $85.17_{\pm0.84}$ | $96.43_{\pm0.36}$ | $81.28_{\pm1.07}$ | | $80.53_{\pm0.72}$ | $\mathbf{100.0}$ | 3.70 |
| | 1-HOP ($\uparrow$) | $72.82_{\pm0.38}$ | $87.92_{\pm1.66}$ | $78.87_{\pm0.91}$ | - | $59.09_{\pm0.49}$ | $\mathbf{96.88}_{\pm0.06}$ | 10.20 |
| | 2-HOP ($\uparrow$) | $65.51_{\pm0.31}$ | $84.41_{\pm2.46}$ | $77.98_{\pm0.95}$ | | $23.09_{\pm0.21}$ | $\mathbf{95.79}_{\pm0.02}$ | 13.49 |
| | 3-HOP ($\uparrow$) | $59.34_{\pm0.62}$ | $80.67_{\pm2.18}$ | $78.65_{\pm0.69}$ | | $58.23_{\pm0.58}$ | $\mathbf{90.19}_{\pm0.22}$ | 11.81 |
| F1 | C2ST-Agg ($\downarrow$) | $95.90_{\pm0.94}$ | $82.52_{\pm0.25}$ | $91.23_{\pm0.39}$ | | $94.55_{\pm0.24}$ | $\mathbf{64.85}_{\pm0.11}$ | 21.41 |
| | Cardinality ($\uparrow$) | $58.17_{\pm3.71}$ | $88.45_{\pm3.05}$ | $56.82_{\pm1.55}$ | | $71.88_{\pm0.12}$ | $\mathbf{100.0}$ | 13.06 |
| | 1-HOP ($\uparrow$) | $77.37_{\pm0.26}$ | $79.35_{\pm0.03}$ | $79.14_{\pm0.72}$ | - | $68.45_{\pm0.20}$ | $\mathbf{93.46}_{\pm0.10}$ | 17.78 |
| | 2-HOP ($\uparrow$) | $76.25_{\pm0.32}$ | $84.18_{\pm0.12}$ | $83.50_{\pm0.82}$ | | $76.93_{\pm0.24}$ | $\mathbf{95.91}_{\pm0.03}$ | 13.94 |
| IMDB | C2ST-Agg ($\downarrow$) | $73.76_{\pm1.78}$ | $65.0_{\pm0.34}$ | $81.56_{\pm2.0}$ | | $53.29_{\pm0.45}$ | | 18.01 |
| | Cardinality ($\uparrow$) | $81.19_{\pm0.80}$ | $98.95_{\pm0.03}$ | $79.53_{\pm1.27}$ | - | DNC | $\mathbf{100.0}$ | 1.06 |
| | 1-HOP ($\uparrow$) | $88.64_{\pm0.70}$ | $91.57_{\pm1.25}$ | $81.76_{\pm0.20}$ | | | $\mathbf{94.84}_{\pm0.35}$ | 3.57 |
| Biodeg. | C2ST-Agg ($\downarrow$) | $88.86_{\pm0.26}$ | | $83.82_{\pm3.35}$ | | $98.02_{\pm0.06}$ | $\mathbf{47.04}_{\pm0.27}$ | 43.88 |
| | Cardinality ($\uparrow$) | $79.53_{\pm0.24}$ | | $85.22_{\pm0.50}$ | | $61.17_{\pm0.36}$ | $\mathbf{100.0}$ | 17.35 |
| | 1-HOP ($\uparrow$) | $61.36_{\pm0.47}$ | - | $75.80_{\pm1.46}$ | - | $49.09_{\pm0.59}$ | $\mathbf{89.37}_{\pm3.76}$ | 17.90 |
| | 2-HOP ($\uparrow$) | $60.54_{\pm0.44}$ | | $77.04_{\pm1.96}$ | | $47.80_{\pm2.16}$ | $\mathbf{86.59}_{\pm5.02}$ | 12.40 |
| Cora | C2ST-Agg ($\downarrow$) | $\mathbf{68.80}_{\pm0.67}$ | | $73.74_{\pm0.47}$ | | $99.59_{\pm0.03}$ | $69.30_{\pm0.52}$ | 0.0 |
| | Cardinality ($\uparrow$) | $96.27_{\pm0.13}$ | - | $90.48_{\pm2.16}$ | - | $68.82_{\pm0.29}$ | $\mathbf{100.0}$ | 3.87 |
| | 1-HOP ($\uparrow$) | $\mathbf{80.42}_{\pm0.34}$ | | $68.39_{\pm0.08}$ | | $4.95_{\pm0.12}$ | $72.16_{\pm1.19}$ | 0.0 |

**Implementation Details.** All reported experiment results are the average of 3 randomly sampled synthetic data samples. Additional implementation details, such as the hardware and software information as well as hyperparameter settings, are in Appendix C.

## 5.2 MULTI-TABLE FIDELITY

We first evaluate multi-table fidelity on the SyntheRela benchmark datasets. We focus on two key metrics: C2ST-Agg, which probes the preservation of higher-order interactions and aggregations across connected tables, and k-hop similarity, which evaluates correlations between columns of paired tables at varying depths within the database schema. In line with previous work, we also report cardinality similarity. However, our approach consistently achieves a perfect score as it preserves the degree distributions. Table 1 shows that RELDIFF consistently outperforms all baselines, securing second-best performance in all but three experiments. Notably, RELDIFF exhibits a significantly smaller performance drop when transitioning from single-table (Table 6) to multi-table C2ST evaluation. Figure 3 illustrates that this degradation is 7× lower than that of the closest competitor, ClavaDDPM, highlighting RELDIFF's superior ability to maintain data fidelity in relational contexts. Additional single-table fidelity and privacy results are in Appendices D.1 and D.3.

Next, we evaluate the results on the ClavaDDPM (Pang et al., 2024) benchmark. Following the original evaluation protocol, we report single-table metrics (Trend and Shape) and multi-table metrics (cardinality and k-hop similarity). We omit the MovieLens and Berka datasets as they are already included in the SyntheRela benchmark. Results in Table 2 show that RELDIFF is the best on all multi-table fidelity metrics, with an average improvement of 25.3% in preserving k-hop correlations. RELDIFF outperforms other methods on all but two single-table evaluations.

To better visualize the fidelity metrics, we plot column-wise marginal densities in appendix D.2.

## 5.3 PERFORMANCE ON DOWNSTREAM TASKS

High-quality synthetic data offers the key advantage of replacing real data for analysis and effective learning on downstream tasks like classification and regression. We evaluate this capacity using Machine Learning Efficiency (ML-E) on RDL tasks (Robinson et al., 2024). According to the RDL

Table 2: **End-to-end results on the ClavaDDPM benchmark**. We follow the evaluation protocol by Pang et al. (2024) and report the cardinality similarity, column shapes, trend scores and correlations between columns in connected tables. DNC denotes *Did Not Converge*.

| | Metric | PrivLava | SDV | ClavaDDPM | RelDiff | Improv. |
|---|---|---|---|---|---|---|
| **California** | Cardinality | 99.90±0.03 | 71.45±0.0 | 99.19±0.29 | **100.0** | 0.10 |
| | Shape | **99.71**±0.02 | 72.32±0.0 | 98.77±0.02 | 99.52±0.02 | 0.0 |
| | Trend | 98.49±0.05 | 50.23±0.0 | 97.65±0.05 | **98.72**±0.01 | 0.24 |
| | 1-HOP | 97.46±0.12 | 54.89±0.0 | 95.16±0.39 | **98.72**±0.0 | 1.29 |
| | AVG 2-WAY | 97.97±0.09 | 52.56±0.0 | 96.41±0.20 | **98.72**±0.01 | 0.77 |
| **Instacart 05** | Cardinality | | | 95.30±0.79 | **100.0** | 4.93 |
| | Shape | | | 89.84±0.29 | **96.85**±0.85 | 7.80 |
| | Trend | DNC | DNC | **99.62**±0.04 | 95.71±0.42 | 0.0 |
| | 1-HOP | | | 76.42±0.39 | **85.83**±1.20 | 12.31 |
| | 2-HOP | | | 39.29±3.38 | **70.74**±0.14 | 80.05 |
| | AVG 2-WAY | | | 76.02±0.78 | **85.78**±0.81 | 12.84 |
| **CCS** | Cardinality | | 74.36±8.40 | 99.25±0.16 | **100.0** | 0.76 |
| | Shape | | 69.04±4.38 | 92.37±2.30 | **98.29**±0.04 | 6.41 |
| | Trend | DNC | 94.84±1.0 | 98.47±0.79 | **98.74**±0.14 | 0.28 |
| | 1-HOP | | 21.74±9.62 | 83.15±4.22 | **89.48**±4.01 | 7.62 |
| | AVG 2-WAY | | 41.68±6.73 | 87.33±3.12 | **92.01**±2.95 | 5.36 |

results presented in Table 3 RELDIFF achieves the best performance on all datasets — sharing the best results on the two datasets with the simplest schema and substantially outperforming other methods on datasets with more complex schemas. This demonstrates our method's competitive capacity to capture and replicate key features of the real data that are most relevant to learning downstream machine learning tasks. We observe that methods with lower performance on data fidelity sometimes outperform stronger methods on utility, highlighting that fidelity and utility are two distinct aspects of synthetic data quality (Hansen et al., 2023). Despite this nuance, RELDIFF consistently achieves strong performance across both aspects.

Table 3: **RDL-utility results.** We report ROC-AUC (higher is better) for classification and MAE (lower is better) for regression tasks. We report the naive baseline scores (mean or majority class) in parentheses. "-" denotes that the utility pipeline could not be used. We **highlight** the best results for each dataset and report the mean and standard error for each metric.

| Dataset | Metric | ORIG. | TabRGN | ClavaDDPM | RCTGAN | REALTABF. | SDV | RelDiff | Improv. |
|---|---|---|---|---|---|---|---|---|---|
| Rossmann | MAE (↓) | 178 (324) | 229±3 | **193**±0.70 | 224±4 | 256±35 | 3428±0.01 | **193**±1 | 0.0 |
| Walmart | MAE (↓) | 9531 (14.7k) | 13848±14 | 11426±1052 | 13435±416 | 13862±300 | 13679±87 | **10475**±1379 | 8.32 |
| Airbnb | AUC (↑) | 0.69 (0.5) | **0.66**±0.01 | 0.51±0.03 | 0.63±0.01 | - | 0.57±0.00 | **0.66**±0.01 | 0.0 |
| Berka | AUC (↑) | 0.81 (0.5) | 0.59±0.23 | 0.52±0.16 | - | - | - | **0.84**±0.02 | 42.4 |
| F1 | AUC (↑) | 0.77 (0.5) | 0.38±0.06 | 0.45±0.06 | 0.48±0.01 | - | 0.52±0.06 | **0.72**±0.01 | 38.5 |

# 6 CONCLUSION

In this work, we introduced RELDIFF, a novel diffusion-based generative framework designed for synthesizing complete relational databases by explicitly modeling their inherent foreign key graph structure. Our approach uniquely combines a joint graph-conditioned diffusion process for attribute synthesis across all inter-connected tables with our $D2K$+SBM graph generator for structure creation. This principled decomposition ensures both high fidelity in the generated data and strict adherence to referential integrity, addressing key limitations of existing relational data synthesis methods that often flatten the relational structure or impose restrictive assumptions.

Through extensive experiments on 11 benchmark datasets, RELDIFF consistently achieves state-of-the-art performance. Our framework effectively captures complex structural and statistical dependencies, leading to synthetic data

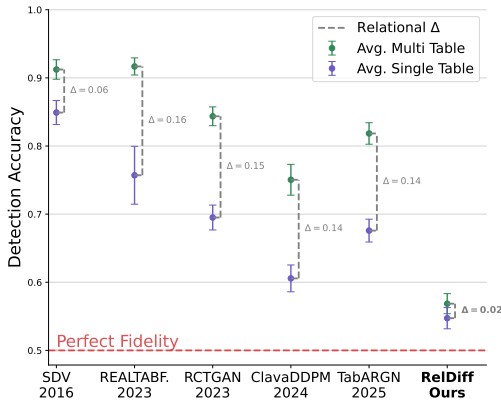

Figure 3: **Comparing single and multi-table C2ST performance**. As opposed to previous methods, our approach incurs only a slight degradation between average multi-table (C2ST-Agg) and single-table (C2ST) performance indicated by the *relational* $\Delta = $ Acc(C2ST-Agg) - Acc(C2ST).

that better reflects the intricacies of real-world relational data. This advancement holds significant promise for various downstream applications, including privacy-preserving data sharing, data augmentation for relational learning tasks, and imputation of missing values in complex relational datasets.

## REPRODUCIBLITY STATEMENT

We provide the full implementation of our method as supplementary material as well as pseudocode of the training procedure in Algorithm 1. Detailed descriptions of the datasets, preprocessing steps, model architecture, training procedure, and evaluation setup are included in Appendix B. Our goal is to ensure full transparency and to facilitate reproducibility in machine learning research.

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

## A  RELATED WORK OVERVIEW

Here we present a more detailed overview of synthetic relational database generation approaches. The Synthetic Data Vault (SDV) (Patki et al., 2016) introduced the first learning-based method for generating relational databases. They introduce a hierarchical modeling approach using Gaussian Copulas, incorporating recursive conditional parameter aggregation to preserve relational structure.

Mami et al. (2022) proposed GraphVAE, a graph-based approach leveraging graph variational autoencoders. They represent relational databases as a single homogeneous graph where all rows become nodes. This contrasts with our approach, which uses a heterogeneous graph to better reflect

the distinct structure and relationships of the table. Foreign keys are implicitly handled by establishing edges between primary and secondary table rows within this homogeneous graph representation. The GraphVAE then uses message-passing layers during both encoding and decoding to model inter-table interactions and generate attribute values for synthetic rows.

Building upon the Conditional Tabular GAN (CT-GAN) method by Xu et al. (2019), two GAN-based methods have been proposed for relational data synthesis. Row Conditional-TGAN (RCTGAN) (Gueye et al., 2023) extends CT-GAN by integrating hierarchical dependencies, enabling the conditional synthesis of child tables based on their parent and grandparent rows. The Incremental Relational Generator (IRG) (Li et al., 2024a) synthesizes relational databases through a sequential table generation process that follows a topological ordering. It constructs an extended table by integrating context from all relevant previously generated or related tables.

The transformer-based approach REaLTabFormer (Solatorio & Dupriez, 2023) focuses on single-parent relational databases. It employs a GPT-2 encoder with a causal language model head to independently model the parent table. For dependent tables, a sequence-to-sequence (Seq2Seq) transformer is utilized, leveraging the frozen parent model as context. All attributes are transformed into a common vocabulary; however, this approach inherits the limitations of language models, particularly concerning the accurate modeling of numerical data.

Xu et al. (2023) introduced a framework for modeling many-to-many datasets using multipartite graphs under differential privacy. Their method utilizes a factorization of the joint data distribution, combining techniques from random graph generation to model structure with graph representation learning methods to conditionally generate tables based on node embeddings. Our approach builds upon their work by specifically capturing hierarchical characteristics during graph generation, modeling tables jointly rather than sequentially, and addressing general relational databases beyond many-to-many relationships.

Diffusion models have also been adapted for relational synthesis. ClavaDDPM (Pang et al., 2024) integrates clustering-guided diffusion models to preserve foreign-key dependencies, utilizing Gaussian mixture models to encode inter-table dependencies. Similarly, RGCLD (Hudovernik, 2024) uses conditional latent diffusion models, using a heterogeneous graph representation and GNNs to encode table relationships, which then guide the diffusion process within the latent space. A key limitation of both methods is their sequential modeling of tables, which introduces implicit assumptions on inter-table dependencies and may allow errors to propagate down the hierarchy during generation.

Finally, auto-regressive models have been explored for tabular and sequential relational synthesis. TabularARGN (Tiwald et al., 2025) employs an any-order auto-regressive network, trained on discretized attributes, to model conditional dependencies. While it specializes in single-table and sequential database modeling, TabularARGN also supports multi-parent schemas by preserving certain dependencies using context tables and maintaining referential integrity for the remaining relationships.

The marginal-based approaches for synthetic relational database generation primarily focus on preserving marginal queries, typically with differential privacy guarantees. PrivLava (Cai et al., 2023) synthesizes relational databases by modeling foreign key relationships as a directed acyclic graph with latent variables, generating tables incrementally. MARE (Kapenekakis et al., 2024), specializing in medical relational data, employs correlation partially directed acyclic graphs (CPDAGs) for selective correlation modeling and orchestrates two-phase data sampling. Alimohammadi et al. (2025) propose an approach to adapt single-table differentially private generators to relational data by learning a weighted bi-adjacency matrix to generate the relational structure. Finally, PrivPetal (Cai et al., 2025) synthesizes a flattened relational database using normalized permutation marginals and then iteratively decomposes it by sampling attributes of reference relations.

We acknowledge two concurrent works GCFM (Scassola et al., 2025) and GRDM (Ketata et al., 2025), which also approach relational data synthesis as a graph-based modeling task. All three methods move beyond sequential table generation to jointly model the database schema using graph-based generative models. While the overarching paradigms are similar, the methods differ in their generative backbones and structural assumptions. Regarding the generative mechanism, GRDM applies a Gaussian DDPM uniformly to both numerical and categorical features, whereas GCFM utilizes flow matching. In terms of structure generation, GRDM introduces a degree-preserving

random graph model, while GCFM utilizes the original graph or samples with replacement from the connected components of the graph.

We classify related work based on how it approaches attribute and structure generation in Table 4.

Table 4: **Classification of related work** based on the mechanisms for attribute and structure modeling as well as the general synthetic data generation (SDG) classification.

| Method | Attribute Modeling | Structure Modeling | SDG Class |
|---|---|---|---|
| SDV (Patki et al., 2016) | conditional tabular | as attributes | statistical |
| GraphVAE (Mami et al., 2022) | homogeneous graph-based | retains original structure | neural |
| RCTGAN (Gueye et al., 2023) | conditional tabular | as attributes | neural |
| IRG (Li et al., 2024a) | conditional tabular | as attributes | neural |
| REaLTabFormer (Solatorio & Dupriez, 2023) | conditional tabular | sequential modeling | neural |
| ClavaDDPM (Pang et al., 2024) | conditional tabular | conditional modeling + matching | neural |
| RGCLD (Hudovernik, 2024) | conditional tabular | retains original structure | neural |
| TabularARGN (Tiwald et al., 2025) | conditional tabular | sequential modeling | neural |
| BayesM2M, NeuralM2M (Xu et al., 2023) | conditional tabular | random graphs (BJDD) | neural & marginal |
| PrivLava (Cai et al., 2023) | conditional tabular | as attributes | marginal |
| MARE (Kapenekakis et al., 2024) | conditional tabular | sequential modeling | marginal |
| DP-Relational (Alimohammadi et al., 2025) | independent tabular | weighted bi-adjacency matrix | marginal |
| PrivPetal (Cai et al., 2025) | flattened tabular | flattening connected tables | marginal |
| Ours | heterogeneous graph-based | $D2K$+SBM random graphs | neural |

# B  DETAILED EXPERIMENT SETUP

This section provides a comprehensive overview of our experimental setup, detailing the datasets (Appendix B.1) utilized and the evaluation metrics (Appendix B.2) employed.

## B.1  DATASETS

Here we describe the datasets used in our evaluation. Table 5 provides detailed statistics for each dataset. The MovieLens and Berka datasets are used in both benchmarks so we only describe them once.

**Rossmann Store Sales**: The Rossmann Store Sales dataset (FlorianKnauer, 2015) features historical sales data for 1115 stores, organized into two connected tables.

**Airbnb**: The Airbnb dataset (Montoya et al., 2015) contains anonymized user interactions and demographics for predicting travel destinations. It comprises multiple tables detailing user sessions and summary information.

**Walmart**: The Walmart dataset (Walmart, 2014) contains historical sales data for 45 stores across three connected tables, including store details, features, and department sales.

**Cora**: The Cora dataset (McCallum et al., 2000) is a graph benchmark of 2708 academic papers classified into seven categories, linked by a citation network of 5429 relationships. Unlike the graph representation learning version with one-hot encoded node features, this relational version stores paper content in a separate table connected via a foreign key, and citation links are represented in a dedicated foreign-key-only table.

**Biodegradability**: The Biodegradability dataset (Blockeel et al., 1999) is a collection of 328 chemical compounds with biodegradation half-life labels, intended for regression analysis based on chemical features.

**IMDB MovieLens**: The IMDB MovieLens dataset (Harper & Konstan, 2015) includes information on movies, actors, directors, user ratings, and related details across seven tables.

**Berka**: The Berka dataset (Berka et al., 2000) is a real-world financial dataset focused on loan outcomes, encompassing loan details and transaction histories across multiple tables. For the SyntheRela benchmark, this dataset is split temporally to facilitate the evaluation of RDL utility.

**F1**: The F1 dataset (F1, 2021) contains historical Formula 1 racing data and statistics from 1950 onwards, covering drivers, races, and results across numerous tables.

**California**: The California dataset is a real-world anonymized census database (M. Center, 2020) on household information. It consists of two tables in the form of a basic parent-child relationship.

**Instacart 05**: The Instacart 05 is created by downsampling 5-percent from the Kaggle competition dataset Instacart (Stanley et al., 2017), which is a real-world transaction dataset of Instacart orders. This dataset consists of 6 tables in total with a maximum depth of 3.

**CCS**: The CCS dataset (Motl & Schulte, 2025) is a real-world transactional dataset Czech debit card company. It consists of 5 tables with a maximum depth of 2.

Table 5: **Summary of the 11 benchmark datasets.** The number of columns represents the number of non-id columns (The MovieLens and Berka datasets appear in both benchmarks). The collection is diverse and covers all types of relational structures.

| Dataset Name | # Tables | # Rows | # Columns | # Relationships | Max Depth | Hierarchy Type |
|---|---|---|---|---|---|---|
| Rossmann | 2 | 59,085 | 16 | 1 | 2 | Linear |
| AirBnB | 2 | 57,217 | 20 | 1 | 2 | Linear |
| Walmart | 3 | 15,317 | 17 | 2 | 2 | Multi Child |
| Cora | 3 | 57,353 | 2 | 3 | 2 | Multi Child |
| Biodegradability | 5 | 21,895 | 6 | 5 | 4 | Multi Child & Parent |
| IMDB MovieLens | 7 | 1,249,411 | 14 | 6 | 2 | Multi Child & Parent |
| Berka | 8 | 757,722 | 37 | 8 | 4 | Multi Child & Parent |
| F1 | 9 | 74,063 | 33 | 13 | 3 | Multi Child & Parent |
| California | 2 | 2,076,141 | 25 | 1 | 2 | Linear |
| CCS | 5 | 423,050 | 11 | 4 | 2 | Multi Child & Parent |
| Instacart 05 | 6 | 1,906,353 | 12 | 6 | 3 | Multi Child & Parent |
| Berka | 8 | 1,079,680 | 41 | 8 | 4 | Multi Child & Parent |
| MovieLens | 7 | 1,249,411 | 14 | 6 | 2 | Multi Child & Parent |

## B.2 METRICS

### B.2.1 SHAPE AND TREND SCORES

Shape and Trend are proposed by the commonly used synthetic data evaluation library SDMetrics[1]. They are used to measure the column-wise density estimation performance and pair-wise column correlation estimation performance, respectively. Shape uses Kolmogorov-Sirnov test (KST) for numerical columns and the total variation distance (TVD) for categorical columns to quantify column-wise density estimation. Trend uses Pearson correlation for numerical columns and contingency similarity for categorical columns to quantify pair-wise correlation.

**Shape**. *KST*: Given two (continuous) distributions $p_r(x)$ and $p_s(x)$ ($r$ denotes real and $s$ denotes synthetic), KST quantifies the distance between the two distributions using the upper bound of the discrepancy between two corresponding cumulative distribution functions (CDFs):

$$\text{KST} = \sup_x |F_r(x) - F_s(x)|, \tag{7}$$

where $F_r(x)$ and $F_s(x)$ are the CDFs of $p_r(x)$ and $p_s(x)$, respectively:

$$F(x) = \int_{-\infty}^{x} p(x)\mathrm{d}x. \tag{8}$$

*TVD* is defined as half the sum of the absolute differences between the real and synthetic probabilities across all categories:

$$\text{TVD} = \frac{1}{2} \sum_{\omega \in \Omega} |R(\omega) - S(\omega)|, \tag{9}$$

where $\omega$ describes all possible categories in a column $\Omega$. $R(\cdot)$ and $S(\cdot)$ denotes the real and synthetic frequencies of these categories. To comply with previous work on relational data synthesis, we report the complement of the KST and TVD distances ($D(P||Q)$) as $1 - D_{\text{KST/TVD}}(P||Q)$

---

[1] https://docs.sdv.dev/sdmetrics

**Trend**. The *Pearson correlation coefficient* measures whether two continuous distributions are linearly correlated and is computed as:

$$\rho_{x,y} = \frac{\text{Cov}(x,y)}{\sigma_x \sigma_y}, \tag{10}$$

where $x$ and $y$ are two continuous columns. Cov is the covariance, and $\sigma$ is the standard deviation.

Then, the performance of correlation estimation is measured by the average differences between the real data correlations and the synthetic data correlations:

$$\text{Pearson Score} = 1 - \frac{1}{2}\mathbb{E}_{x,y}\Big[|\rho^R(x,y) - \rho^S(x,y)|\Big], \tag{11}$$

where $\rho^R(x,y)$ and $\rho^S(x,y)$ denotes the Pearson correlation coefficient between column $x$ and column $y$ of the real data and synthetic data, respectively. As $\rho \in [-1,1]$, the average distance of Pearson coefficients is divided by 2, to ensure that it falls in the range of $[0,1]$, and subtracted from 1 such that the larger the score, the better the estimation.

*Contingency similarity*: For a pair of categorical columns $A$ and $B$, the contingency similarity score computes the difference between the contingency tables using the TVD. The process is summarized by the formula below:

$$\text{Contingency Score} = 1 - \frac{1}{2}\sum_{\alpha \in A}\sum_{\beta \in B}|R_{\alpha,\beta} - S_{\alpha,\beta}|, \tag{12}$$

where $\alpha$ and $\beta$ describe all the possible categories in column $A$ and column $B$, respectively. $R_{\alpha,\beta}$ and $S_{\alpha,\beta}$ are the joint frequency of $\alpha$ and $\beta$ in the real data and synthetic data, respectively. To obtain a score, the distance is again subtracted from 1.

### B.2.2 K-HOP TREND

The k-hop Trend metric, proposed by (Pang et al., 2024), extends the Trend metric to evaluate the preservation of correlations across multiple tables connected by foreign keys. The 0-hop Trend is equivalent to the standard Trend metric, measuring pairwise column correlations within a single table. For $k > 0$, the k-hop Trend assesses correlations across tables reachable within $k$ foreign key hops. This is achieved through a series of join operations:

- 1-hop: Refers to the correlation between two columns in tables that are directly linked by a foreign key. To calculate this, we join the two related tables based on the foreign key and then compute the Trend metric on the resulting joined table.

- $(k > 1)$-hop: Extends the process iteratively. Based on a foreign key sequence of length $k$, the tables are recursively joined. The Trend metric is then computed on the final joined table.

As in the Trend metric, the Pearson correlation coefficient and contingency similarity are used to quantify the differences in joint distributions depending on the data type. The final k-hop Trend score is the average of these correlation/similarity scores across all relevant k-hop relationships within the database schema. Similar to the 0-hop Trend, a higher score indicates a better preservation of inter-table correlations up to $k$ hops.

### B.2.3 CARDINALITY SIMILARITY

Given a parent table $P$ with a primary key $pk$ and a child table $C$ with a foreign key $fk$ referencing $pk$, this metric evaluates the similarity of the cardinality distribution between the real and synthetic datasets. The cardinality of a parent row $p \in P$ is defined as the number of child rows $c \in C$ for which $c.fk = p.pk$.

Let $card_R(p)$ and $card_S(p)$ denote the cardinality of a parent row $p$ in the real and synthetic datasets, respectively. This yields two numerical distributions: $D_R = \{card_R(p) \mid p \in P_{\text{real}}\}$ and $D_S = \{card_S(p) \mid p \in P_{\text{synthetic}}\}$.

The cardinality similarity score is computed as the complement of the Kolmogorov-Smirnov statistic, defined as $1 - \text{KST}(D_R, D_S)$, where KST refers to the Kolmogorov-Smirnov Test statistic as defined in Appendix B.2.1. Cardinality similarity ranges from $0$ to $1$, where a score of $1$ indicates identical cardinality distributions in the real and synthetic data, and a score of $0$ indicates maximally different distributions.

### B.2.4  C2ST

The Classifier Two-Sample Test (C2ST) evaluates the fidelity of synthetic data by training a discriminator (in our case, an XGBoost model) to distinguish it from real data. This detection-based approach, rooted in two-sample testing, uses the classifier's performance as a proxy for distributional similarity. If the discriminator achieves better-than-random accuracy, it indicates discernible differences between the real and synthetic datasets, suggesting lower fidelity. C2ST offers a comprehensive assessment of single table fidelity that captures complex higher-order dependencies between features beyond simple correlations when using expressive-enough models, as highlighted in (Zein & Urvoy, 2022).

### B.2.5  C2ST-Agg

The Classifier Two-Sample Test with Aggregations (C2ST-Agg) (Jurkovič et al., 2025) extends detection-based fidelity evaluation to relational data by capturing inter-table relationships. C2ST-Agg functions by augmenting parent tables with aggregated features derived from their connected child tables. This propositionalization approach, drawing inspiration from relational reasoning techniques, enables the C2ST to evaluate the preservation of interactions between columns in connected tables, as well as relationship cardinalities. Conceptually, C2ST-Agg assesses how well fundamental SQL operations such as join and groupby are maintained alongside complex interactions both within and between tables. By summarizing child-table information using aggregation functions (e.g., mean, count, max), C2ST-Agg effectively accounts for both relationship cardinality and high-level interactions across related tables, offering a comprehensive assessment of relational data fidelity.

In our evaluation we use the aggregation functions used by the original authors *mean* - for numerical attributes, *count* of connected rows and *count distinct* for categorical variables and use an XGBoost (Chen & Guestrin, 2016) model with k=5 fold cross-validation as the discriminative model.

### B.2.6  RDL Utility

The relational deep learning utility (RDL utility) (Jurkovič et al., 2025) metric evaluates the capacity of synthetic relational data to support effective learning on downstream tasks. This metric adheres to the RelBench (Robinson et al., 2024) framework, which transforms relational databases into temporal heterogeneous graphs with predefined tasks specifically designed for GNN training.

To assess utility, the RDL utility metric employs the RelBench GNN pipeline. A heterogeneous variant of the GraphSage model (Hamilton et al., 2017) is trained on both the real and synthetic data. These trained models are subsequently evaluated on a dedicated test set composed entirely of real data.

Data splitting is handled via a time-based splitting strategy. Our evaluation incorporates five datasets from the SyntheRela benchmark that possess a temporal feature, with the following predictive tasks:

- Rossmann: Prediction of daily number of customers for each store and date.
- Walmart: Prediction of weekly sales for each department within each store.
- Airbnb: Binary prediction of whether a user has previously made a booking.
- Berka: Prediction of the binary loan status (successful or unsuccessful).
- F1: Prediction of whether a driver will qualify in the top-3 for a race in the next month.

All datasets and models utilize the default RelBench hyperparameters.

### B.2.7  Distance to Closest Record

The distance to closest record (DCR) (Zhao et al., 2021) evaluates privacy protection by measuring how closely synthetic data resembles the training data. The DCR score (Zhang et al., 2024) quantifies

the fraction of synthetic records whose nearest neighbor is in the training set; a value near 0.5 suggests the model samples from the true distribution rather than overfitting.

Nearest neighbor distances are calculated using the $l^2$ norm of synthetic, training, and holdout records. For a synthetic record $i$, distances to its nearest neighbors in training ($N_{\text{trn}}$) and holdout ($N_{\text{hold}}$) datasets are:

$$\text{d(i)}_{\text{trn}} = \min_{j \in N_{\text{trn}}} \|\text{syn}_i - \text{trn}_j\|_2, \quad \text{d(i)}_{\text{hold}} = \min_{j \in N_{\text{hold}}} \|\text{syn}_i - \text{hold}_j\|_2.$$

An indicator function, $I(i)_{\text{trn}}$, determines if the nearest neighbor of synthetic record $i$ is in the training set:

$$I(i)_{\text{trn}} = \begin{cases} 1 & \text{if } \text{d(i)}_{\text{trn}} < \text{d(i)}_{\text{hold}}, \\ 0 & \text{if } \text{d(i)}_{\text{trn}} > \text{d(i)}_{\text{hold}}, \\ 0.5 & \text{if } \text{d(i)}_{\text{trn}} = \text{d(i)}_{\text{hold}}. \end{cases}$$

The DCR score is then computed as:

$$\text{DCR score} = \frac{1}{N_{\text{syn}}} \sum_{i=1}^{N_{\text{syn}}} I(i)_{\text{trn}}.$$

### B.3 MEMBERSHIP INFERENCE ATTACKS

Membership inference attacks (MIA) are used to evaluate the privacy risk of a synthetic data generator. An MIA measures the vulnerability of a model by assessing whether an adversary, with access to both the synthetic data and real data from the same population, can infer if a specific record from the real data was used in the training process of the synthetic data generator (El Emam et al., 2022). Our evaluation uses the implementation provided by Lautrup et al. (2025), which assumes a scenario where the adversary knows the population but has no prior knowledge of the synthesis algorithm. We model the attacker as a LightGBM classifier (Shi et al., 2025b) trained to distinguish between synthetic records and real records from an external holdout dataset. The classifier's performance in identifying whether a sample is real or synthetic serves as a proxy for the risk of membership disclosure. We report both the *recall* and *precision* of the attacking model, where recall is analogous to the membership inference risk. Similarly as with the DCR metric, we evaluate MIA at the individual table level, as no multi-table implementation currently exists.

## C   IMPLEMENTATION DETAILS

We implemented RELDIFF in PyTorch. We performed our experiments on two Nvidia H100 GPUs with 80G memory.

**Data preprocessing.** Raw relational datasets often contain missing values. Our initial preprocessing step involves imputing these, following approaches in (Pang et al., 2024; Shi et al., 2025a): numerical missing values are replaced by the column average, and categorical missing values are treated as a distinct new category. For the SyntheRela benchmark, we adopt a more nuanced approach for missing values, similar to (Patki et al., 2016; Hudovernik, 2024). We introduce an additional binary indicator variable for each attribute to explicitly denote whether a value was originally missing; this indicator is modeled as a separate categorical variable, allowing us to recover the original missingness pattern after sampling. To mitigate training instability caused by the diverse ranges of numerical columns, we transform the numerical values with the QuantileTransformer[2] and recover the original values after sampling.

**Hyperparameters Setting.** RELDIFF employs a consistent hyperparameter setting across all datasets, with the sole exception of the number of epochs and batch size, which is mainly dependent on the graph structure. We train our models for 10000 epochs on most datasets. For larger network datasets, specifically *Instacart 05* and *MovieLens*, we utilize a reduced number of epochs (400 and 4000 respectively) to manage computational load. We use the AdamW optimizer with learning rate $\gamma = 6e-4$ and weight decay $w = 1e-5$ in all experiments.

---

[2]https://scikit-learn.org/stable/modules/generated/sklearn.preprocessing.QuantileTransformer.html

Regarding the specific hyperparameters within RELDIFF, the values for $\sigma_{\min}$ and $\sigma_{\max}$ are set to 0.002 and 80.0, respectively, referencing the optimal setting in (Karras et al., 2022). The parameter $\delta$ is set to $1e-3$. For the loss weightings, we fix $\lambda_{\mathrm{cat}}$ to 1.0 and linearly decay $\lambda_{\mathrm{num}}$ from 1.0 to 0.0 as training proceeds. In all our experiments, the number of GNN layers is set to $k = 2$. During inference, we select the checkpoint with the lowest training loss. And utilize 100 discretization steps ($T = 100$) during sampling.

**Model Architecture** Our model is parameterized by a heterogeneous graph neural network, featuring transformer encoders and decoders and an MLP backbone. Each column is initially projected into a $d$-dimensional vector using a linear layer, with $d = 4$, matching the size used in Zhang et al. (2024) and Shi et al. (2025a). These tokenized columns are then processed by a two-layer transformer. Subsequently, the concatenated columns are projected to a dimension of $dim_h = 128$, to which noise embeddings of the same dimensionality are added, consistent with the approach of Pang et al. (2024).

The embeddings are then processed by a heterogeneous variant of the GraphSAGE network (Hamilton et al., 2017), which is used as the RDL baseline in Fey et al. (2024). For databases containing records at fixed time intervals, and given our use of a permutation-invariant GNN, positional encodings are added to the embeddings before message passing to preserve record order. The GNN embeddings for each table are then further processed by five-layer MLPs, conditioned on a time embedding. The size of these MLPs for each table is comparable to those used in experiments by Kotelnikov et al. (2023). Finally, the hidden representation is decoded back into the data space by another two-layer transformer. It is worth noting that the MLP backbone accounts for the majority of the model's parameters, and the memory consumed by these parameters is typically less than that used by the intermediate data representations, especially since relational databases often contain larger tables than typical tabular datasets.

# D ADDITIONAL EXPERIMENTS

## D.1 SINGLE TABLE FIDELITY RESULTS

Table 6: **Single-table results**. For each dataset and metric we report the average detection accuracy (C2ST - lower is better), column shapes and column pair trends (Shape, Trend - higher is better) across all tables for three independent samples. DNC denotes *Did Not Converge* and "-" denotes a method is unable to generate the dataset. The best result is **bolded**. We report the percentage improvement of RELDIFF over the state-of-the-art in blue.

| | Metric | TabARGN | ClavaDDPM | RCTGAN | REALTABF. | SDV | RelDiff | Improv. |
|---|---|---|---|---|---|---|---|---|
| Airbnb | C2ST ($\downarrow$) | $64.23_{\pm0.20}$ | $78.10_{\pm0.03}$ | $88.37_{\pm0.14}$ | $83.97_{\pm4.36}$ | $99.75_{\pm5e\text{-}3}$ | $\mathbf{54.11_{\pm0.34}}$ | 15.76 |
| | Shape ($\uparrow$) | $95.70_{\pm0.05}$ | $94.42_{\pm0.01}$ | $89.18_{\pm0.17}$ | $71.66_{\pm0.92}$ | $59.37_{\pm0.04}$ | $\mathbf{98.14_{\pm0.07}}$ | 2.55 |
| | Trend ($\uparrow$) | $93.48_{\pm0.33}$ | $87.78_{\pm0.12}$ | $79.37_{\pm0.29}$ | $53.90_{\pm1.26}$ | $49.03_{\pm0.08}$ | $\mathbf{95.76_{\pm0.25}}$ | 2.43 |
| Rossmann | C2ST ($\downarrow$) | $56.07_{\pm0.58}$ | $66.77_{\pm0.14}$ | $88.02_{\pm0.50}$ | $74.70_{\pm1.55}$ | $96.90_{\pm0.21}$ | $\mathbf{52.46_{\pm0.32}}$ | 6.44 |
| | Shape ($\uparrow$) | $96.96_{\pm0.19}$ | $94.05_{\pm0.07}$ | $91.31_{\pm0.04}$ | $90.65_{\pm0.38}$ | $81.05_{\pm0.19}$ | $\mathbf{98.04_{\pm0.07}}$ | 1.12 |
| | Trend ($\uparrow$) | $91.34_{\pm0.08}$ | $84.78_{\pm0.80}$ | $84.38_{\pm0.40}$ | $84.58_{\pm0.88}$ | $67.77_{\pm0.25}$ | $\mathbf{95.93_{\pm0.65}}$ | 5.02 |
| Walmart | C2ST ($\downarrow$) | $83.54_{\pm0.84}$ | $\mathbf{53.50_{\pm1.95}}$ | $76.40_{\pm0.55}$ | $70.87_{\pm1.07}$ | $87.02_{\pm0.81}$ | $60.30_{\pm0.50}$ | 0.0 |
| | Shape ($\uparrow$) | $89.09_{\pm0.33}$ | $92.21_{\pm0.52}$ | $82.31_{\pm0.51}$ | $81.71_{\pm0.40}$ | $81.80_{\pm0.10}$ | $\mathbf{94.04_{\pm0.53}}$ | 1.99 |
| | Trend ($\uparrow$) | $83.89_{\pm0.21}$ | $94.02_{\pm0.14}$ | $86.60_{\pm0.25}$ | $83.10_{\pm0.46}$ | $87.61_{\pm0.23}$ | $\mathbf{95.42_{\pm0.45}}$ | 1.49 |
| Berka | C2ST ($\downarrow$) | $72.31_{\pm0.17}$ | $54.48_{\pm0.11}$ | $68.12_{\pm0.44}$ | | $82.40_{\pm0.33}$ | $\mathbf{50.23_{\pm0.05}}$ | 7.80 |
| | Shape ($\uparrow$) | $82.20_{\pm0.26}$ | $91.62_{\pm0.10}$ | $81.90_{\pm0.38}$ | - | $56.27_{\pm0.29}$ | $\mathbf{97.72_{\pm0.03}}$ | 6.65 |
| | Trend ($\uparrow$) | $70.43_{\pm0.25}$ | $88.54_{\pm1.19}$ | $74.22_{\pm0.28}$ | | $64.01_{\pm0.11}$ | $\mathbf{98.81_{\pm0.02}}$ | 11.59 |
| F1 | C2ST ($\downarrow$) | $81.93_{\pm0.49}$ | $71.42_{\pm0.46}$ | $80.67_{\pm0.31}$ | | $89.84_{\pm0.22}$ | $\mathbf{63.50_{\pm0.03}}$ | 11.09 |
| | Shape ($\uparrow$) | $84.71_{\pm1.15}$ | $84.63_{\pm0.28}$ | $89.68_{\pm0.40}$ | - | $52.62_{\pm0.57}$ | $\mathbf{94.89_{\pm0.08}}$ | 5.81 |
| | Trend ($\uparrow$) | $81.31_{\pm0.45}$ | $84.65_{\pm0.05}$ | $90.17_{\pm0.03}$ | | $73.05_{\pm0.19}$ | $\mathbf{95.10_{\pm0.12}}$ | 5.46 |
| IMDB | C2ST ($\downarrow$) | $50.92_{\pm0.22}$ | $\mathbf{49.83_{\pm0.07}}$ | $55.38_{\pm0.11}$ | | | $52.07_{\pm0.36}$ | 0.0 |
| | Shape ($\uparrow$) | $98.40_{\pm0.14}$ | $\mathbf{99.01_{\pm0.05}}$ | $92.70_{\pm0.09}$ | - | DNC | $96.91_{\pm0.49}$ | 0.0 |
| | Trend ($\uparrow$) | $97.80_{\pm0.13}$ | $\mathbf{98.66_{\pm0.10}}$ | $81.65_{\pm0.03}$ | | | $93.88_{\pm0.87}$ | 0.0 |
| Biodeg. | C2ST ($\downarrow$) | $58.79_{\pm0.19}$ | | $58.26_{\pm0.15}$ | | $68.59_{\pm0.14}$ | $\mathbf{48.28_{\pm0.20}}$ | 17.13 |
| | Shape ($\uparrow$) | $90.85_{\pm0.14}$ | - | $90.91_{\pm0.40}$ | - | $79.46_{\pm0.47}$ | $\mathbf{95.95_{\pm0.11}}$ | 5.55 |
| | Trend ($\uparrow$) | $74.82_{\pm0.35}$ | | $85.44_{\pm2.19}$ | | $97.58_{\pm0.50}$ | $\mathbf{99.38_{\pm0.13}}$ | 1.85 |
| Cora | C2ST ($\downarrow$) | $50.94_{\pm0.37}$ | | $\mathbf{48.97_{\pm0.14}}$ | | $75.45_{\pm0.16}$ | $54.03_{\pm0.61}$ | 0.0 |
| | Shape ($\uparrow$) | $92.65_{\pm0.12}$ | - | $\mathbf{96.38_{\pm0.15}}$ | - | $50.24_{\pm0.17}$ | $87.85_{\pm1.22}$ | 0.0 |

In this section, we present detailed single-table fidelity results on the SyntheRela benchmark. We evaluate the Shape, Trend, and C2ST scores. As shown in Table 6, RELDIFF consistently demonstrates the strongest performance. When employing the original structure, RELDIFF is only outperformed on the IMDB and Cora datasets, indicating a trade-off where some single-table fidelity is exchanged for superior multi-table fidelity. This behavior on the IMDB dataset can be attributed to structural motifs that may cause bottlenecks in GNN message passing. Notably, our approach can achieve near-perfect performance on Cora if its schema is normalized to third normal form.

## D.2   VISUALIZATION OF GENERATED DATA

To complement our quantitative evaluation and offer a qualitative evaluation of generation quality, we provide visual comparisons between the real and synthetic data distributions.

Figure 4 presents the column-wise feature densities for selected tables. We utilize kernel density estimation (KDE) curves for numerical features and histograms for categorical features. The plots demonstrate that the marginal densities produced by RELDIFF consistently align closely with the real data distributions, visually corroborating RELDIFF's superior performance on the Shape metric.

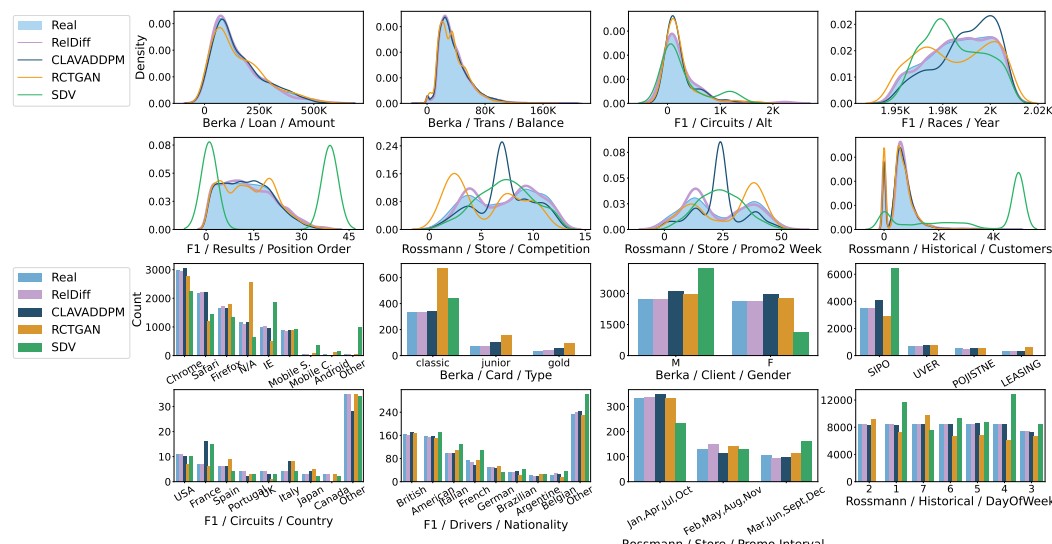

Figure 4: **Visualizations of column-wise densities.** The horizontal axis labels stand for "<Dataset> / <Table> / <Feature>".

In Figure 5, we visualize the pairwise column correlations within single tables. As illustrated by the plots, RELDIFF faithfully captures the correlations in the data, exhibiting a higher degree of fidelity compared to the baseline methods.

Finally, we visualize long-range dependencies in Figure 6. These plots highlight RELDIFF's ability to preserve statistical interactions between columns across connected tables, validating the effectiveness of our joint graph-based diffusion process.

## D.3   PRIVACY EVALUATION

**Distance to Closest Record (DCR).** We follow related work (Kotelnikov et al., 2023; Zhang et al., 2024; Shi et al., 2025a; Pang et al., 2024) to perform a privacy sanity check against SMOTE (Chawla et al., 2002), an interpolation-based method that generates new data through convex combinations of real data points. To quantify the privacy level, we evaluate the distance to closest record (Zhao et al., 2021). Specifically, we compare the DCR distributions of RELDIFF against SMOTE on two datasets, adhering to the evaluation protocol of Pang et al. (2024): *California*, a real-world census dataset containing anonymized household and individual information, and a subset of tables from the *Berka* dataset, which holds anonymized financial information from a Czech bank. The results of the DCR score (Zhang et al., 2024) are presented in Table 7.

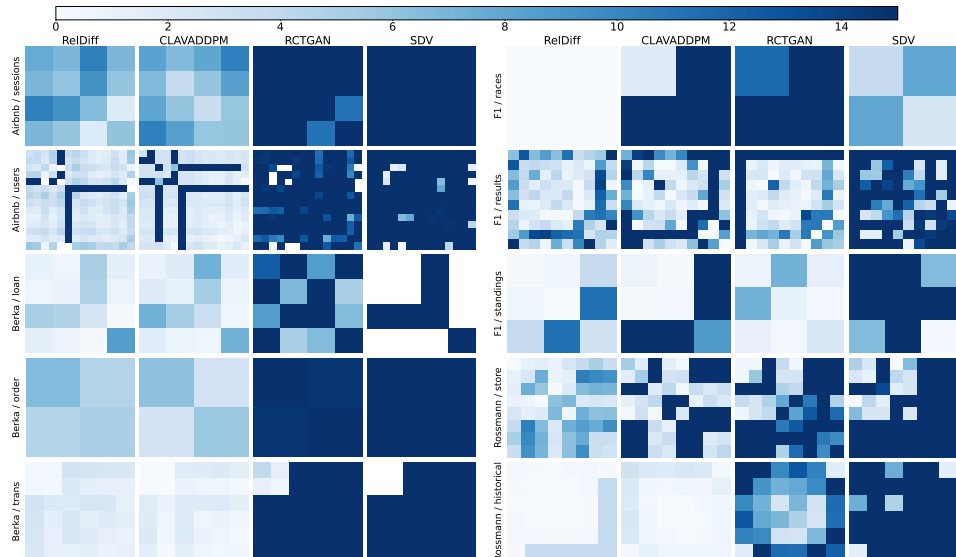

Figure 5: **Visualizations of single-table column-wise correlations.** The shades represents the error rate (i.e., difference in Trend metric), and the vertical axis labels stand for "<Dataset> / <Table>".

Table 7: **DCR score** represents the probability that a synthetic example is closer to the training set rather than the holdout set (%, a score closer to 50% is better).

| Method Table | Household | Individual | Transaction | Order |
|---|---|---|---|---|
| SMOTE | 77.22 ±0.0 | 76.25 ±0.0 | 99.94 ±0.0 | 99.40 ±0.0 |
| ClavaDDPM | **50.23** ±0.0 | **50.31** ±0.0 | 53.47 ±0.0 | 91.84 ±0.0 |
| **RelDiff** | 50.54 ±0.0 | 50.45 ±0.0 | **50.71** ±0.0 | **52.38** ±0.0 |

RELDIFF consistently achieves DCR scores around 50%. This outcome is indicative of the model's ability to sample from the underlying data distribution rather than memorizing the training data.

**Membership Inference Attack (MIA).**

The results of the membership inference attack (MIA) are summarized in Table 8, which reports the *Macro F1 score* of an adversarial classifier. The Macro F1 aggregates precision and recall across both the member and non-member classes, providing a balanced measure of the attack's effectiveness even under class imbalance (a score of 50% corresponds to random guessing).

Table 8: **MIA Macro F1** denotes the effectiveness of an adversarial classifier in distinguishing training members from non-members. We report means with standard errors; values near or below 50% indicate performance near random guessing and thus limited privacy risk.

| Method Table | Household | Individual | Transaction | Order |
|---|---|---|---|---|
| SMOTE | 50.99 ±0.05 | 50.59 ±0.05 | 63.23 ±0.21 | 59.28 ±0.62 |
| CLAVADDPM | 49.59 ±0.07 | 50.08 ±0.04 | 51.70 ±0.20 | 55.89 ±0.35 |
| **RelDiff** | **48.81** ±0.14 | **48.7** ±0.07 | **49.01** ±0.05 | **51.55** ±0.35 |

In principle, if a generative model were perfectly private, the expected F1 score of an attacker would be close to 50%. In our experiments, the observed values cluster tightly around this baseline and are consistently lower than those of competing methods, indicating limited membership leakage. On three tables, the F1 scores are even slightly below 50%, which can be explained by the presence of a small fraction of outlier values generated by our model. Since such values never occur in the original data, the attacker learns to separate them from the holdout distribution, but this signal is not informative for true membership inference. Overall, our privacy evaluation shows strong empirical privacy protection.

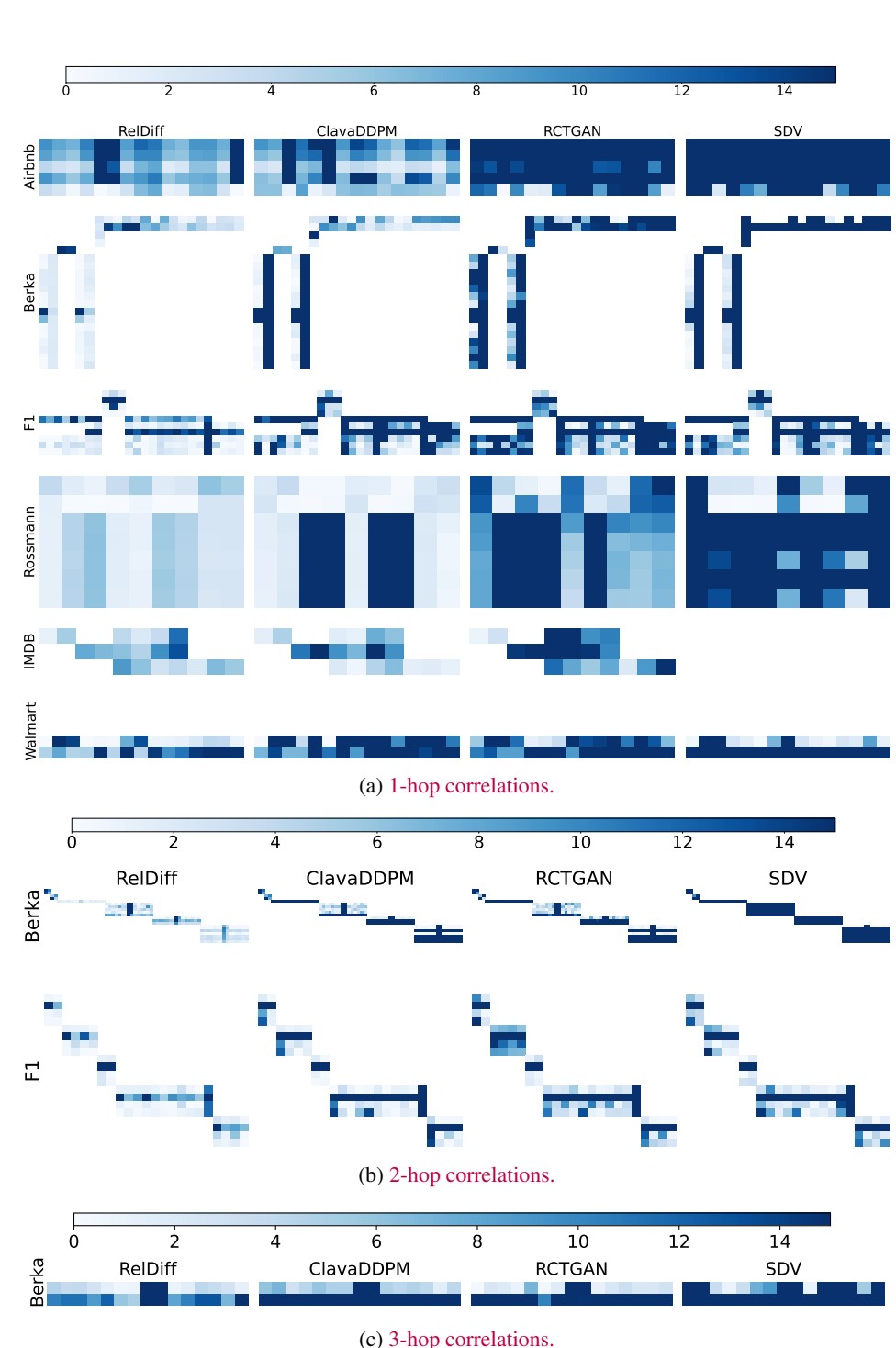

Figure 6: **Visualizations of k-hop correlations** for $k \in \{1, 2, 3\}$. The shades represent the error rate (i.e., difference in k-hop Trend metrics between columns in tables at distance $k$).

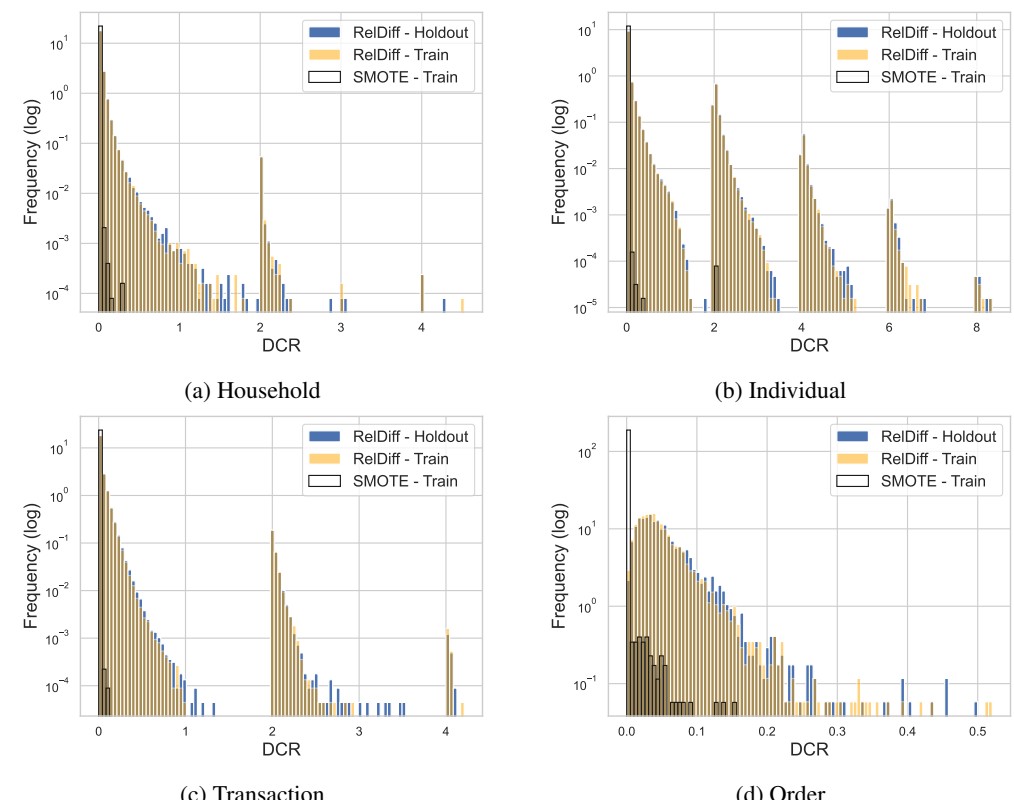

(a) Household

(b) Individual

(c) Transaction

(d) Order

Figure 7: **DCR distributions** on the *California* and *Berka* datasets (log-transformed y-axis). RELDIFF exhibits DCR values for the training set that are significantly higher than SMOTE, indicating enhanced privacy protection. The distribution of DCR values for the held-out data remains consistent with that of the training data.

## D.4 ABLATION STUDIES

We conducted an ablation study to evaluate the contribution of key components of our method. The results are summarized in Table 9.

**Graph Generation Strategy.** We conducted a series of ablations on our graph generation approach. We compared our full method, which uses the $D2K$+SBM generator, against two baselines: a simpler bipartite $2K$ model and our method when provided with the original ground-truth graph (*original* column). For the simple two-table Airbnb dataset, both graph generators perfectly recover the original graph structure. However, as schema complexity increases for the Berka and F1 datasets, the effectiveness of our $D2K$+SBM generator becomes evident, as it significantly outperforms the bipartite $2K$ variant. This highlights the importance of our approach for accurately modeling complex relational structures.

**Model Architectures.** Next, we ablate the GNN architectures to examine their importance in capturing inter-table relationships. When we remove the GNN entirely, the method reverts to a single-table generative model with graph based sampling of rows and preserved relational structure. While this variant preserves the distributions of individual columns, it fails to model long-range inter-table relationships, as evidenced by the significant performance drop on all HOP metrics. This confirms the GNN's central role in our approach. We also evaluated a more complex, heterogeneous variant of the Graph Attention Network (GAT) as a replacement for our GNN. The results were mixed, with performance improving on some datasets while decreasing on others. Given that the GAT significantly increases runtime, we conclude that our basic GNN offers a sufficient and efficient trade-off between performance and efficiency for our joint diffusion approach. We also ablate the transformer layers. Contrary to previous work (Shi et al., 2025a), we observed that removing the

Table 9: **Ablation Study** on three databases with increasingly complex schemas. We present results for our method when trained with the ground-truth entity graph (RELDIFF original), our proposed RELDIFF $D2K$+SBM variant, a version with a more complex GAT network (RELDIFF GAT), and variants where the transformer layers or GNN are removed entirely. Additionally, we include a variant that uses a simpler bipartite 2K entity graph generator, as well as how sampling parameters, such as number of neighbors (S) and hops (K), affect performance. For each metric, we highlight the best-performing variant in **bold** and underline the second-best.

| | Metric | RelDiff (original) | RelDiff Bipartite 2K | RelDiff D2K+SBM | RelDiff GAT | RelDiff w/o Transformer | RelDiff S=64 | RelDiff K=1 | RelDiff w/o GNN |
|---|---|---|---|---|---|---|---|---|---|
| Airbnb | Shape (↑) | $98.14_{\pm0.07}$ | $98.14_{\pm0.07}$ | $98.14_{\pm0.07}$ | $\underline{98.44}_{\pm0.03}$ | $98.40_{\pm0.05}$ | $98.40_{\pm0.08}$ | $98.02_{\pm0.04}$ | $\mathbf{99.06}_{\pm0.05}$ |
| | Trend (↑) | $95.76_{\pm0.25}$ | $95.76_{\pm0.25}$ | $95.76_{\pm0.25}$ | $\underline{95.92}_{\pm0.04}$ | $\mathbf{96.46}_{\pm0.20}$ | $95.54_{\pm0.36}$ | $95.12_{\pm0.31}$ | $95.89_{\pm0.12}$ |
| | 1-HOP (↑) | $89.37_{\pm0.38}$ | $\mathbf{89.37}_{\pm0.38}$ | $89.37_{\pm0.38}$ | $86.63_{\pm0.69}$ | $\underline{88.66}_{\pm0.33}$ | $87.72_{\pm0.29}$ | $86.97_{\pm0.37}$ | $84.65_{\pm0.11}$ |
| Berka | Shape (↑) | $\mathbf{98.75}_{\pm8e\text{-}4}$ | $97.94_{\pm0.06}$ | $97.92_{\pm0.01}$ | $96.79_{\pm0.06}$ | $97.94_{\pm0.02}$ | $95.07_{\pm0.02}$ | $97.83_{\pm0.09}$ | $\underline{98.72}_{\pm0.03}$ |
| | Trend (↑) | $\mathbf{99.35}_{\pm5e\text{-}3}$ | $98.70_{\pm0.01}$ | $98.68_{\pm0.02}$ | $98.01_{\pm0.01}$ | $98.77_{\pm0.04}$ | $97.40_{\pm0.01}$ | $98.91_{\pm0.01}$ | $\underline{98.97}_{\pm0.04}$ |
| | 1-HOP (↑) | $\mathbf{98.03}_{\pm0.06}$ | $96.59_{\pm0.06}$ | $\underline{96.70}_{\pm0.06}$ | $94.83_{\pm0.01}$ | $96.53_{\pm0.05}$ | $94.01_{\pm0.06}$ | $92.48_{\pm0.08}$ | $89.94_{\pm0.43}$ |
| | 2-HOP (↑) | $\mathbf{97.96}_{\pm0.03}$ | $95.33_{\pm0.03}$ | $\underline{95.61}_{\pm0.03}$ | $93.03_{\pm0.01}$ | $95.00_{\pm0.09}$ | $95.59_{\pm0.04}$ | $95.59_{\pm0.05}$ | $88.45_{\pm0.78}$ |
| | 3-HOP (↑) | $\underline{91.97}_{\pm0.13}$ | $90.43_{\pm0.22}$ | $89.54_{\pm0.38}$ | $90.33_{\pm0.33}$ | $90.95_{\pm0.15}$ | $\mathbf{93.20}_{\pm0.17}$ | $90.41_{\pm0.12}$ | $85.26_{\pm0.87}$ |
| F1 | Shape (↑) | $\mathbf{97.92}_{\pm0.05}$ | $82.80_{\pm0.16}$ | $\underline{96.80}_{\pm0.03}$ | $92.95_{\pm0.36}$ | $96.69_{\pm0.14}$ | $96.66_{\pm0.07}$ | $93.94_{\pm0.17}$ | $96.13_{\pm0.05}$ |
| | Trend (↑) | $\mathbf{98.04}_{\pm0.01}$ | $89.35_{\pm0.13}$ | $97.35_{\pm0.05}$ | $95.86_{\pm0.07}$ | $\underline{97.51}_{\pm0.07}$ | $96.50_{\pm0.01}$ | $94.88_{\pm0.06}$ | $95.88_{\pm0.02}$ |
| | 1-HOP (↑) | $\mathbf{95.10}_{\pm0.14}$ | $84.48_{\pm0.28}$ | $\underline{94.02}_{\pm0.11}$ | $86.18_{\pm0.83}$ | $93.55_{\pm0.06}$ | $93.84_{\pm0.20}$ | $90.02_{\pm0.46}$ | $79.40_{\pm0.64}$ |
| | 2-HOP (↑) | $\mathbf{97.92}_{\pm0.05}$ | $91.26_{\pm0.11}$ | $96.19_{\pm0.01}$ | $96.07_{\pm0.04}$ | $96.22_{\pm0.03}$ | $\underline{97.73}_{\pm0.03}$ | $95.14_{\pm0.11}$ | $83.43_{\pm0.42}$ |

transformer encoders and decoders can lead to slight performance improvements on certain metrics. This suggests that users can opt for a simplified architecture without the transformer layers. On tables with a high number of columns, this might even result in significant speedups due to the quadratic complexity of attention.

**Subgraph Sampling.** Finally, we investigate the sensitivity of our model to subgraph sampling parameters, specifically the number of sampled neighbors ($S$) and the number of GNN layers ($K$). By default, we sample all neighbors and use $K = 2$. When limiting the neighborhood size to $S = 64$, we observe that the method retains the majority of its performance across metrics. Surprisingly, this constraint even leads to improved preservation of long-range dependencies in certain cases, such as the 3 and 2 HOP metrics on the Berka and F1 datasets respectively. Additionally, reducing the receptive field to a single hop ($K = 1$) results in only a marginal degradation in fidelity while offering substantial computational benefits. We attribute this effectiveness to the multi-step nature of the diffusion process, where information propagates iteratively, effectively expanding the receptive field to $\mathcal{O}(T \times K)$ hops over $T$ denoising steps. Consequently, $K = 1$ successfully captures long-range dependencies while offering significant computational gains, reducing training and generation times by over 40% on the Berka dataset. These findings suggest that neighborhood subsampling and shallower architectures can be effectively leveraged to further enhance scalability on massive databases with minimal impact on generation quality.

# E STUDY OF MODEL EFFICIENCY AND SCALABILITY

In this section, we analyze the efficiency and scalability of RELDIFF. We provide a comparative complexity analysis against alternative graph generation approaches and evaluate runtime performance relative to the state-of-the-art baseline ClavaDDPM. Both training and sampling runtimes are reported. Finally, we study RELDIFF 's scaleability by generating (i) a database with an order of magnitude more rows than those considered in prior work and (ii) a database whose schema contains an order of magnitude more tables.

## E.1 COMPLEXITY ANALYSIS

**Graph Generation.**

We compare the computational complexity of our proposed $D2K$+SBM generator against common deep learning–based graph generative approaches. Let $N$ denote the number of nodes in the relational entity graph. Transformer-based autoregressive models incur $\mathcal{O}(N^4)$ complexity, since each generation step requires full self-attention across all nodes and sequentially generates the adjacency matrix. This quickly becomes intractable as $N$ grows. Diffusion-based graph generators reduce this cost but

still require $\mathcal{O}(TN^2)$ operations, where $T$ is the number of diffusion steps. At each denoising step, dense pairwise computations across nodes are performed, leading to quadratic cost per step.

In contrast, our $D2K$+SBM generator achieves $\mathcal{O}\left(ln^2(N) + \sum_{i=1}^{|\mathcal{R}|}(N + 2e_i + B)\right)$ complexity, where $|\mathcal{R}| \ll N$ is the number of foreign key relationships, $e_i$ the number of edges per relation, $E = \sum_{i=1}^{|\mathcal{R}|} e_i$ the total number of edges, and $B < N$ the number of blocks in the stochastic block model. This formulation yields near-linear scaling with the number of nodes and edges in the graph, in sharp contrast to the polynomial scaling of deep learning–based methods. If we opt to transform *two-parent-no-child* nodes into many-to-many edges we further improve efficiency as we decrease the number of nodes and edges in the graph.

Moreover, while existing graph generators require an additional post-processing step to enforce referential integrity, our method guarantees it by construction. Taken together, this analysis demonstrates the superior scalability of $D2K$+SBM: whereas autoregressive and diffusion-based approaches become prohibitively expensive on large graphs, our generator remains efficient, making it particularly well-suited for synthesizing large relational databases with many entities and complex schemas.

**Data Generation.** We analyze the computational complexity of our joint diffusion model, comparing its computational efficiency against its closest competitor, ClavaDDPM, during both training and inference phases. The training complexity of both RELDIFF and ClavaDDPM is primarily dictated by their diffusion-based backbones, which have a comparable number of parameters. The key difference in efficiency emerges from their respective denoising conditioning mechanisms.

ClavaDDPM conditions the sampling process with a separate MLP classifier for each relationship within the database schema. Consequently, its complexity scales linearly with the number of relationships. This approach becomes computationally expensive as the complexity of the schema increases.

In contrast, RELDIFF employs a single heterogeneous GNN. The GNN's per-batch complexity is $O\left(\prod_{i=1}^{K} S_i\right)$, controlled by the structure of local neighborhoods with small, bounded depth $K = 2$, and sparse node degrees or explicit sampling thresholds $S_i$ (Hamilton et al., 2017). This allows RELDIFF to maintain superior efficiency as the database schema becomes more complex.

The inference complexity is proportional to the training complexity. Again, ClavaDDPM repeats denoising steps for tables with multiple parents, whereas RELDIFF performs a single forward pass per diffusion step across the entire relational graph. This significantly minimizes redundant computations and enhances efficiency as the schmea grows more complex. Additionally, RELDIFF employs more efficient diffusion modeling, which allows it to generate high-quality samples with 1/20 of the steps required by DDPM-based methods. This allows our method to achieves significant runtime improvements compared to ClavaDDPM.

### E.2  RUNTIMES

We now report the training and sampling runtimes of RELDIFF and ClavaDDPM, measured across the benchmark datasets used in our experiments. As shown in Table 10, RELDIFF achieves training runtimes that are competitive with, and frequently faster than, ClavaDDPM. Notably, RELDIFF is significantly more efficient on the complex F1 schema (2.7h vs. 12.3h). While training times are higher on Berka and IMDB due to our default full-neighbor sampling strategy on these denser graphs, our ablation studies (Appendix D.4) demonstrate that users can alleviate this cost by subsampling neighbors, achieving speedups of up to 30% with minimal impact on fidelity. Furthermore, we emphasize that training represents a one-time cost. In practice, sampling will be the more commonly executed operation, where RELDIFF achieves significantly lower runtimes.

Table 11 highlights this key advantage, showing that RELDIFF is significantly faster during sampling. For instance, on the California dataset, RELDIFF is more than 20 times faster, a direct result of our method's ability to jointly generate data, avoiding the need for sequential generation. This finding is particularly important as sampling is the process that will be executed most frequently in a production environment.

We report the maximum GPU memory usage during training and the sampling and training batch sizes in Table 12.

Table 10: **Training runtimes** (hours).

| Dataset | RelDiff | ClavaDDPM |
|---|---|---|
| California | 3.3 | 3.3 |
| CCS | 2.5 | 3.2 |
| Instacart 05 | 4.8 | 6.6 |
| Airbnb | 0.4 | 0.4 |
| Rossman | 0.4 | 1.2 |
| Walmart | 0.6 | 0.9 |
| Berka | 11.8 | 6.5 |
| IMDB | 7.7 | 3.1 |
| F1 | 2.7 | 12.3 |
| Biodegradability | 0.3 | - |
| CORA | 0.5 | - |

Table 11: **Sampling runtimes** (seconds).

| Dataset | RelDiff | ClavaDDPM |
|---|---|---|
| California | 221.2 | 5,032 |
| CCS | 126.3 | 1,232 |
| Instacart 05 | 5,107 | 9,968 |
| Airbnb | 8.6 | 178.5 |
| Rossman | 5.7 | 194.9 |
| Walmart | 17.8 | 40.2 |
| Berka | 780.6 | 1,662 |
| IMDB | 499.4 | 2,314 |
| F1 | 32.3 | 643.6 |
| Biodegradability | 8.4 | - |
| CORA | 9.4 | - |

Table 12: **Memory usage and batch size**.

| Dataset | Training | | Sampling |
| | Batch size | Max Mem. Usage | Batch size |
|---|---|---|---|
| California | 50k | 42 GB | 100k |
| CCS | 100k | 19 GB | 200k |
| Instacart 05 | 40k | 45 GB | 80k |
| Airbnb | Full | 6 GB | Full |
| Rossman | 2048 | 7 GB | Full |
| Walmart | 512 | 7 GB | Full |
| Berka | 50k | 21 GB | 100k |
| IMDB | 200k | 34 GB | 400k |
| F1 | 20k | 5 GB | Full |
| Biodegradability | 512 | 3 GB | Full |
| CORA | 25k | 7 GB | 50k |

### E.3 LARGE-SCALE DATABASE GENERATION

We perform additional experiments on large-scale databases to further validate the scalability of our method. The loss curves in Figure 8 demonstrate that RELDIFF successfully trains on the large rel-hm dataset, proving its capability to handle millions of rows within a reasonable training time (under 48 hours). Similarly, the stable loss curve on the highly complex AdventureWorks schema confirms that our graph-based approach can effectively model databases with a large number of interconnected tables, a crucial test for its applicability to real-world, enterprise-level data. This suggests that future methods should now be evaluated not just on small-to-medium datasets but on large-scale databases with millions of rows and highly complex schemas (See Figure 9 for the AdventureWorks schema).

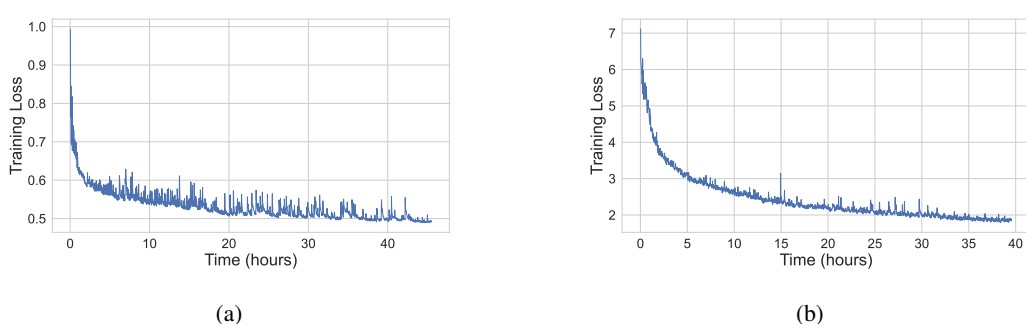

Figure 8: Loss curves for the *rel-hm* (a) and *AdventureWorks* (b) datasets.

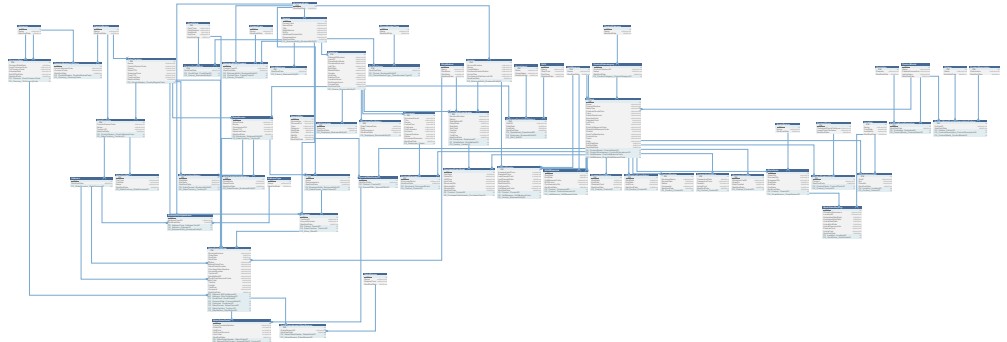

Figure 9: Schema of the AdventureWorks database.

## F MODELING THE JOINT DISTRIBUTION $p(\mathcal{V}, \mathcal{E})$

To construct a general, data-agnostic generative framework, we formulate the synthesis of relational databases as sampling from the joint distribution over entities and edges of the relational entity graph (Section 3), $p(\mathcal{V}, \mathcal{E})$. An ideal simulation of the data generating process might attempt to mimic the temporal evolution of a database, sequentially adding users, products, and transactions. However, such an approach requires extensive, dataset-specific domain knowledge. Consequently, we seek a factorization of the joint distribution that remains computationally tractable without relying on specific assumptions about the data's origin. We identify three viable factorizations: (1) direct joint modeling $p(\mathcal{V}, \mathcal{E})$, (2) an attributes-first decomposition $p(\mathcal{V})p(\mathcal{E}|\mathcal{V})$, and (3) a structure-first decomposition $p(\mathcal{E})p(\mathcal{V}|\mathcal{E})$.

We posit that the first two approaches face a fundamental scalability bottleneck. In both direct joint modeling and features-first generation, the edges are generated alongside or subsequent to the entity attributes. This necessitates evaluating the probability of a link existing between every possible pair of entities. For a relational database containing tables of size $N$ and $M$, this implies modeling interactions over a search space of size $N \times M$ (See Appendix E.1). This quadratic complexity renders factorizations (A) and (B) computationally prohibitive for large-scale real-world databases, where the number of potential interactions grows exponentially with the number of tables.

In contrast, the structure-first factorization $p(\mathcal{E})p(\mathcal{V}|\mathcal{E})$ effectively circumvents this combinatorial explosion. By generating the relational skeleton first, we can exploit the exchangeability of nodes and utilize efficient random graph models—such as the D2K+SBM proposed in this work that scales linearly with the number of edges rather than quadratically with the number of nodes. While the dependency between structure and attributes is essentially bidirectional, generating the lower-dimensional topology first provides a stable computational graph for the subsequent attribute generation. Crucially, this factorization does not assume that only the attributes can depend on structure. Our parameterization of $p(\mathcal{V}|\mathcal{E})$ employs an expressive GNN model that jointly models attributes across the entire schema. This enables the model to capture bidirectional dependencies between node attributes and structure, effectively filling in the graph with attributes that respect the underlying distribution. For

example the model can learn to assign a subset of values that would result in the row having few foreign key connections, to a node with the corresponding structure in the graph. This is analogous to autoregressive language modeling, where an imposed sequential generation order does not preclude the model from capturing complex, non-causal semantic dependencies between tokens.

## G  BROADER IMPACTS

This research introduces a novel method for generating synthetic relational databases, which presents potential benefits in fields with privacy restrictions, such as healthcare, finance, and education, and in scenarios involving limited or biased data. However, there are potential negative impacts to consider. While our empirical privacy analysis does not raise immediate concerns, our method is not equipped with provable privacy guarantees like differential privacy. Additionally, due to the method's ability to generate datasets that closely resemble the original data, it might inadvertently amplify biases already present in the original data. Furthermore, synthetic data that closely mirrors real data could be misused. Consequently, we believe that future work should prioritize research directions focused on enhancing privacy protection and developing effective bias reduction techniques for synthetic relational data.

