# OpenReview forum: "RelDiff: Relational Data Generative Modeling with Graph-Based Diffusion Models"
_ICLR.cc/2026/Conference — Submitted to ICLR 2026_

### Official Review · Reviewer_5xxZ · 2025-10-19

**Soundness:** 3
**Presentation:** 3
**Contribution:** 2
**Rating:** 4
**Confidence:** 4

**Summary:**

This paper introduces RELDIFF, a generative framework for synthesizing complex relational databases. Unlike prior methods that flatten schemas or assume conditional independence, RELDIFF explicitly models database structures as graphs and uses a graph-based diffusion model to generate mixed-type attributes across interconnected tables. The framework ensures referential integrity via a D2K + SBM graph generator and captures both inter- and intra-table dependencies using GNNs. Experiments are conducted on 11 datasets.

**Strengths:**

1.The paper is generally well-written and easy to follow.

2.The use of the D2K + SBM graph generator to preserve foreign key cardinality and hierarchical dependencies is novel and technically interesting.

3.The ablation study is comprehensive.

**Weaknesses:**

1. The decomposition $p(\mathcal{V},\mathcal{E})$ = $p(\mathcal{E})p(\mathcal{V}|\mathcal{E})$ is assumed without theoretical support.

2. The proposed joint diffusion model is not clearly novel compared with existing tabular diffusion approaches such as TabDDPM, TABSYN, and TabDiff.

3. The high training cost of RelDiff raises scalability concerns, and memory usage is not reported.

**Questions:**

1. The statement “tabular data includes complex and varied distributions” (lines 41-42) appears somewhat vague. Image and text datasets can also exhibit diverse and complex distributions due to varying sources and contexts. Could the authors clarify in what specific sense tabular data distributions are considered more complex or varied?

2. The decomposition $p(\mathcal{V},\mathcal{E})$ = $p(\mathcal{E})p(\mathcal{V}|\mathcal{E})$ seems to be taken as a modeling assumption without sufficient justification. It is unclear why the generative process is assumed to first sample the relational structure and then the attributes. In practice, foreign-key relationships (edges) may be influenced by attribute distributions (e.g., business logic or temporal constraints), while attribute distributions can also be constrained by the structure (e.g., table hierarchy and connection density). Therefore, this factorization implicitly assumes a unidirectional dependency from structure generation to attribute generation, yet the paper provides neither theoretical justification nor empirical evidence to support this assumption.

3. The proposed joint diffusion model seems not an innovative design. The use of diffusion models for generating heterogeneous tabular features (i.e., numerical and categorical) has been extensively studied in prior works such as TabDDPM [1], TABSYN [2], and TabDiff [3]. The authors are encouraged to clarify what makes their proposed hybrid generation method novel beyond existing tabular diffusion approaches and to provide stronger empirical evidence demonstrating its superior effectiveness.

4. While quantitative metrics are provided, the quality of the generated tabular data should be further demonstrated through visualization to offer more intuitive and interpretable evidence of the model’s effectiveness.

5. As shown in Table 10, the training cost of RelDiff is substantially higher than that of ClavaDDPM, raising concerns about the method’s scalability and practicality on large-scale datasets. Moreover, the paper does not report the memory cost across different datasets, which is important for assessing the overall efficiency and deployability of the proposed framework.

[1] TabDDPM: Modeling tabular data with diffusion models. ICML2023

[2]  Mixed-type tabular data synthesis with score-based diffusion in latent space. ICLR2024

[3] TabDiff: a Mixed-type Diffusion Model for Tabular Data Generation. ICLR2025

---

> ### Author Response · Authors · 2025-11-27
>
> > Q1: The statement “tabular data includes complex and varied distributions” (lines 41-42) appears somewhat vague.
>
> **Q1)** By "complex and varied," we refer specifically to the unique statistical heterogeneity of tabular data compared to homogeneous modalities like images or text. Unlike images (continuous pixels) or text (tokens from a shared vocabulary), tabular data mixes continuous numerical values with categorical features with potentially different cardinalities, requiring the generative model to handle fundamentally different probability distributions simultaneously. It is also not uncommon for tabular data to include imbalanced distributions and missing values, further adding to the complexity.
> To clarify this, we have replaced the vague phrasing with *heterogeneous and often imbalanced distributions*.
>
> > Q4: While quantitative metrics are provided, the quality of the generated tabular data should be further demonstrated through visualization to offer more intuitive and interpretable evidence of the model’s effectiveness.
>
> **Q4)** We thank the reviewer for this suggestion. We have added visualizations of the generated data to Appendix D, demonstrating RelDiff’s superior generative ability when compared to prior works.
>
> **W1, Q2) On the Factorization of the Joint Distribution**
> The reviewer would like to see our justification for our choice of factorizing the joint distribution and is concerned that it assumes a unidirectional dependency from structure generation to attribute generation.
>
> We agree that real-world dependencies between attributes and structure are often bidirectional. However, the decomposition $p(\mathcal{V}, \mathcal{E}) = p(\mathcal{E})p(\mathcal{V}|\mathcal{E})$ is a necessary design choice driven by computational tractability rather than a disregard for these dependencies.
>
> We have added a discussion on alternative factorizations (such as generating attributes first) in Appendix F and added an explanation and reference to it in Section 4.1. In short, as mentioned in Appendix E.1, generating edges with deep learning methods suffer from a combinatorial explosion, as they require evaluating potential links between every pair of entities ($O(N^2)$ complexity). In contrast, our "structure-first" approach allows us to exploit node exchangeability and utilize efficient random graph models (D2K+SBM) scaling near linearly with the number of nodes and edges.This  allows us to generate edges in linear time $O(N+E)$, which is the only tractable approach for large-scale database synthesis.
>
> Crucially, this factorization does **not** necessitate a  that only attributes depend on structure. By first establishing the topology and then modeling $p(\mathcal{V}|\mathcal{E})$ with an expressive graph neural network, our model "fills in" attributes that are statistically consistent with the generated structure. This is analogous to large language models, which can successfully capture non-causal semantic dependencies despite relying on a strictly sequential generation order.

---

> > ### Author Response · Authors · 2025-11-27
> >
> > **W2, Q3) Comparison to Single-Table Diffusion Models**
> > The reviewer asks us to clarify the difference bewteen our method and prior work for single-table generation.
> > We clarify that while RelDiff utilizes established diffusion processes for the marginal distributions of mixed-type data (similar to Shi et al., 2025, that build upon [1,2]), it introduces a fundamentally different generative paradigm compared to tabular diffusion models (TabDDPM, TabSyn):
> >
> > **Graph-Based vs. I.I.D. Modeling:** Tabular diffusion models assume rows are independent and identically distributed (i.i.d.). When used to model a relational database, tabular models treat tables as isolated marginals and disregard the relational structure. In contrast, RELDIFF is a graph-based diffusion model. Jointly denoising attributes across nodes in the graph for all tables simultaneously. This allows it to model interactions between tables that standard tabular models are not able to. The reason that the optimization objective is similar to tabular diffusion model is the fact that we utilize the same forward process for different attributes and the fact that we weight the loss contributions of individual tables proportionally to the number of rows in the table to account for cases with highly imbalanced row counts across tables.
> >
> > **Joint vs. Sequential Synthesis:** Unlike prior relational methods [3, 4, 5, 6]  that treat synthesis as a sequence of conditional tasks, (hence introducing limiting assumptions), RELDIFF models the joint distribution of the entire database. The reverse process is fundamentally different: it models the joint distributions of nodes across the entire schema, rather than optimizing for individual table’s conditional distributions.
> >
> > **Intractability of the Alternative:** Using standard tabular diffusion models to generate tables independently would necessitate a subsequent step to synthesize the edges conditionally on the attributes. As we characterize in Appendix E.1, generating edges while accounting for all table attributes using deep models is computationally prohibitive due to quadratic complexity with respect to the number of rows. RELDIFF’s joint formulation offers a tractable way to capture these interactions using diffusion (as explained in response to W1).
> >
> > We have added a comparison with the state of the art tabular method TabDiff with randomly generated structure to demonstrate that tabular methods are unable to preserve multi-table fidelity. Our results align with those reported in previous work, where Pang et al [3] compare their multi-table method to TabDDPM.
> >
> > | Dataset | Metric | RelDiff  | TabDiff |
> > | :--- | :--- | :--- | :--- |
> > | **Airbnb** | 1-HOP ($\uparrow$) | **89.37 $\pm$ 0.38** | 84.29 $\pm$ 1.4 |
> > | **F1** | 1-HOP ($\uparrow$) | **94.02 $\pm$ 0.1** | 80.23 $\pm$ 1.5 |
> > | | 2-HOP ($\uparrow$) | **96.19 $\pm$ 0.0** | 85.85 $\pm$ 1.9 |
> > | **Berka** | 1-HOP ($\uparrow$) | **96.70 $\pm$ 0.0** | 83.52 $\pm$ 1.7 |
> > | | 2-HOP ($\uparrow$) | **95.61 $\pm$ 0.03** | 80.21 $\pm$ 1.1 |
> > | | 3-HOP ($\uparrow$) | **89.54 $\pm$ 0.38** | 77.67 $\pm$ 3.6 |
> >
> > We have also updated our discussion in Appendix D.4 to clarify the distinction between RelDiff and single-table methods.
> >
> > [1] Sahoo et al. "Simple and effective masked diffusion language models." (2024).
> >
> > [2] Karras, Tero, et al. "Elucidating the design space of diffusion-based generative models." (2022).
> >
> > [3] Pang, et al. "Clavaddpm: Multi-relational data synthesis with cluster-guided diffusion models." (2024).
> >
> > [4] Solatorio and Dupriez. "Realtabformer: Generating realistic relational and tabular data using transformers." (2023).
> >
> > [5] Hudovernik. “Relational data generation with graph neural networks and latent diffusion models.” (2024).
> >
> > [6] Tiwald, et al. "Tabularargn: A flexible and efficient auto-regressive framework for generating high-fidelity synthetic data." (2025).

---

> ### Author Response · Authors · 2025-11-27
>
> **W3, Q5) Regarding Scalability**
> The reviewer is concerned about the long training times of our method. We appreciate the reviewer's concerns as it has helped us significantly improve the methods usability. We acknowledge that the training times initially reported were high; however, this was primarily due to unoptimized training setup rather than inherent algorithmic limitations.
>
> **1. Optimization and Speedup** By decreasing the number of training epochs (to 5K) and implementing standard engineering optimizations—specifically model compilation, removing unnecessary device synchronizations, and utilizing mixed-precision training—we have achieved a **5x speedup** in training times. For instance, the training time for the California dataset was reduced to 3.3 hours (down from 17.4 hours) without any drop in generation quality. This effectively matches the training duration of ClavaDDPM while retaining our significant advantage in sampling efficiency, where RelDiff is over **20x** faster.
>
>
> **2. Practical Implications** We emphasize that training is a one-time cost, whereas sampling—where RelDiff excels in efficiency—is the frequent operation in deployment. Regarding memory, all experiments were conducted on a single GPU. Peak memory usage for the reported numbers on the California dataset was  42GB during the optimized run. As shown in Appendix E.3, our method successfully scales to databases significantly larger and more complex than those manageable by prior methods on a single GPU.
>
> We are re-running experiments across all datasets with these optimizations. We will update Table 10 in the final revision to reflect the improved training times and also report the corresponding memory usage.
>
> Additionally, as demonstrated by our extended ablation studies based on input from Reviewer 1, the framework allows users to tune hyperparameters to further improve runtime with minimal impact on fidelity:
> - Reducing Hops ($K=1$): Can yield up to a 45% speedup while retaining ~95% of performance.
> - Neighbor Subsampling: Offers up to a 20% speedup.
> Removing Transformer Encoders:
> - Provides up to a 15% speedup.

---

> > ### Author Response · Authors · 2025-12-04
> > **Update on Scalability Results**
> >
> > As promised, we have completed the re-evaluation of our training efficiency with the optimized codebase and have updated **Table 10** and added **Table 12**  to the manuscript. The results confirm that our optimizations have resolved the initial scalability concerns:
> > - **Training Runtimes**: RelDiff is now faster than or equal to ClavaDDPM on 7 out of the 9 datasets compared. Notably, we achieve substantial speedups on complex schemas, such as **F1 (2.7h vs 12.3h, $\approx$4.5x faster)**. While training remains slower on the dense Berka and IMDB graphs due to full-neighborhood sampling, our ablation results confirm that subsampling can reduce this cost by up to 40% with minimal fidelity loss, matching the ClavaDDPM runtimes.
> > - **Memory Usage**: We have added **Table 12** reporting the peak memory consumption for all datasets. The maximum memory usage observed was **45 GB** (on Instacart), with the majority of datasets requiring significantly less. This confirms that RelDiff can be comfortably trained on a single modern high-end GPU (e.g., NVIDIA A100 or A6000), demonstrating its deployability without requiring complex multi-GPU infrastructure.

---

### Official Review · Reviewer_6VFj · 2025-10-30

**Soundness:** 3
**Presentation:** 3
**Contribution:** 2
**Rating:** 4
**Confidence:** 4

**Summary:**

The paper tackles relational data generation via graph diffusion that first resamples a D2K+ SBM foreign-key graph to preserve referential integrity, and then jointly denoises numerical and categorical attributes using a heterogeneous GNN, achieving state-of-the-art multi-table fidelity and up to an 80\% improvement in k-hop correlations across 11 real-world databases.

**Strengths:**

**Quality**. The paper uses a combination of different techniques. First, it generates a graph via their D2K + SBM generator. Their generator is comprised of Bayesian SBM as a model of graphs + D2K graph generator to preserve relationships between nodes. Subsequently, they define a conditional hybrid diffusion process which generates categorical and numerical samples conditioned on the generated graph.

**Clarity**. Paper is easy to follow.

**Significance**. Paper looks at tabular data generation for relational databases.

**Originality**. A conditional generation framework of integrating graphs could be interesting to the community.

**Weaknesses:**

Overall, experiments and ablation studies are comprehensive, comprising of performance, runtime, computation and privacy.  However, a concern I have is its novelty. Its a combination of existing well-known methods which I believe for the current standards of conferences like NeurIPS, ICLR and ICML, it may be insufficient. The main takeaway that the framework provides is that integrating graph based generators into diffusion models help provide extra signal to improve generative performance.

**Questions:**

Please see weaknesses.

---

> ### Author Response · Authors · 2025-11-27
>
> We thank the reviewer for recognizing the comprehensive nature of our experiments and ablation studies. However, we respectfully disagree with the characterization of RelDiff as merely a combination of existing methods or that its main contribution is simply providing "extra signal" via graph generators. Our method represents a fundamentally different approach to relational database synthesis, moving from sequential autoregressive modeling to joint graph-based diffusion. We do not merely integrate graph based generators but define a joint diffusion process of the entire database. We highlight two key aspects of our method’s novelty:
>
> 1. **First Joint Modeling Framework**: Prior state-of-the-art methods [1,2,3,4] treat relational synthesis as a sequence of conditional table generation tasks ($p(T_2|T_1)p(T_1)$). This imposes strict ordering assumptions and suffers from error propagation down the table hierarchy. RelDiff is the first framework (alongside concurrent work [5, 6]) to model the database as a single heterogeneous graph, jointly denoising attributes across all tables simultaneously.
>
> 2. **Handling Arbitrarily Complex Schemas**: The reviewer suggests our graph generator only provides "extra signal." In reality, our principled decomposition of structure ($p(\mathcal{E})$ via SBM) and attributes ($p(\mathcal{V}|\mathcal{E})$ via joint diffusion) allows RelDiff to efficiently synthesize databases with arbitrarily complex schemas—including cyclic dependencies and many-to-many relationships—which sequential methods are unable to model. This capability is not an incremental improvement but a key advantage, allowing our method to synthesize databases with schemas orders of magnitude more complex than previously used in related work (as demonstrated in Appendix E.3), while preserving the referential integrity by design. We have added the schema of the AdventureWorks database to the Appendix to further emphasize this point (Figure 9 in the updated manuscript).
>
> We believe the shift from sequential table generation to joint graph diffusion is a critical step forward for the field, offering capabilities (schema flexibility, joint modeling of tables) that "combinations of existing methods" do not possess.
>
> [1] Pang, et al. "Clavaddpm: Multi-relational data synthesis with cluster-guided diffusion models." (2024).
>
> [2] Solatorio and Dupriez. "Realtabformer: Generating realistic relational and tabular data using transformers." (2023).
>
> [3] Hudovernik. “Relational data generation with graph neural networks and latent diffusion models.” (2024).
>
> [4] Tiwald, et al. "Tabularargn: A flexible and efficient auto-regressive framework for generating high-fidelity synthetic data." (2025).
>
> [5] Ketata, et al. "Joint Relational Database Generation via Graph-Conditional Diffusion Models." (2025).
>
> [6] Scassola,et al. "Graph Conditional Flow Matching for Relational Data Generation." (2025).

---

### Official Review · Reviewer_ghnQ · 2025-11-01

**Soundness:** 2
**Presentation:** 2
**Contribution:** 2
**Rating:** 6
**Confidence:** 2

**Summary:**

This paper tackles synthetic relational database generation. Rather than flattening multi-table schemas or generating tables in a pre-set order, the authors decompose the task into sampling a relational entity graph that respects foreign-key cardinalities and hierarchy using a microcanonical, degree-corrected Stochastic Block Model, and a joint, graph-conditioned diffusion model that denoises mixed-type attributes across all tables with a heterogeneous GNN. Training uses subgraph neighbor sampling and a hybrid continuous + categorical masking diffusion objective; sampling first draws a new entity graph with the SBM module, then jointly denoises attributes conditioned on the graph. On two benchmarks covering 11 real-world databases, the method reports stronger multi-table fidelity and good downstream RDL utility compared to ClavaDDPM, RCTGAN, SDV, RealTabFormer, TabularARGN, and PrivLava.

**Strengths:**

1. The modeling choices are well-motivated: microcanonical SBM gives hard constraints for referential integrity; hybrid diffusion aligns with mixed continuous/categorical columns; heterogeneous GNNs with subgraph sampling are a sensible scalability strategy.
2. Joint graph-conditioned diffusion over the entire entity graph, coupled with a microcanonical, nested SBM to preserve relational cardinalities and hierarchy, is a clean and compelling synthesis.

**Weaknesses:**

1. The baselines omit recent joint modeling approaches like GRDM (Graph-Conditional Relational Diffusion Model), which also performs joint denoising over relational graphs and reports strong k-hop performance. The paper positions prior work mainly as sequential/conditional (ClavaDDPM, etc.), but the landscape now includes joint graph-conditioned diffusion and flow-matching variants.
2. The nested SBM is well-motivated for modular schemas, but the paper preprocesses two-parent/no-child tables to many-to-many edges and then learns blocks and degrees under hard constraints. That can bias structure when schemas are weakly modular and may interact with functional dependencies in ways the SBM cannot express.

**Questions:**

1. How sensitive are results to the neighborhood radius and the subgraph neighbor-sampling scheme during training and inference?
2. Many databases encode temporal edges. Can RelDiff model time-stamped relations and reproduce temporal integrity?

---

> ### Author Response · Authors · 2025-11-27
>
> **W1) Comparison with Concurrent Work:** The reviewer points out two recent works that propose a similar generative models as our own. We first highlight that these are concurrent work (preprints were released at the same time as ours and were not yet published at the time of our submission). Nevertheless, to address the reviewers concerns we have added a reference and comparison to these works to our related work section in Appendix A. In summary, both methods approach relational data generation as a graph-based modeling task, the key differences are in (i) how they construct their diffusion models [1] use a Gaussian DDPM for both numerical and categorical features while [2] utilize flow matching; (ii) and how they approach graph generation [1] proposes a degree-preserving random graph model and [2] retains the structure of the original graph.  Additionally, our method accounts for dimension tables, and missing values during generation.
>
> Upon the reviewers' request, we have also included a comparison of the methods. GRDM has not yet released their code publicly, so we use the same setup used in their paper.
>
> | Dataset        | Metric             | RelDiff (Ours)       | GCFM                | GRDM                |
> |----------------|--------------------|----------------------|---------------------|---------------------|
> | **California** | Shape ($\uparrow$) | **99.52 $\pm$ 0.01** | 97.96 $\pm$ 0.06     | 99.15 $\pm$ 0.01    |
> |                | Trend ($\uparrow$) | **98.65 $\pm$ 0.02** | 95.90 $\pm$ 0.03     | 98.02 $\pm$ 0.02    |
> |                | 1-HOP ($\uparrow$) | **98.70 $\pm$ 0.03** | 95.26 $\pm$ 0.15     | 97.68 $\pm$ 0.02    |
> | **IMDB**       | Shape              | 97.72 $\pm$ 0.20     | 97.83 $\pm$ 0.26     | **98.25 $\pm$ 0.1** |
> |                | Trend              | 95.61 $\pm$ 0.21     | **96.72 $\pm$ 0.42** | 96.31 $\pm$ 0.15    |
> |                | 1-HOP ($\uparrow$) | **96.47 $\pm$ 0.03** | 92.42 $\pm$ 1.58    | 94.19 $\pm$ 2.12    |
> | **Berka**      | Shape              | **98.34 $\pm$ 0.34** | 90.39 $\pm$ 0.85    | 97.00 $\pm$ 0.14    |
> |                | Trend              | **98.82 $\pm$ 0.41** | 86.68 $\pm$ 0.80    | 98.24 $\pm$ 0.06    |
> |                | 1-HOP ($\uparrow$) | **97.58 $\pm$ 0.51** | 86.47 $\pm$ 0.74    | 93.16 $\pm$ 0.02    |
> |                | 2-HOP ($\uparrow$) | **97.73 $\pm$ 0.61** | 89.69 $\pm$ 0.56    | 96.67 $\pm$ 0.14    |
> |                | 3-HOP ($\uparrow$) | 91.41 $\pm$ 0.65 | 87.85 $\pm$ 0.64    | **94.47 $\pm$ 0.04**    |
>
>
> RelDiff achieves the best performance on 8 out of 11 metrics, consistently outperforming concurrent approaches on California and Berka in preserving both single-table distributions (Shape, Trend) and direct correlations (1-hop, 2-hop). While GRDM shows a slight advantage in 3-hop correlations on Berka, we attribute this to the trade-off for sampling efficiency. RelDiff uses 10x fewer denoising steps than GRDM giving the model fewer steps to propagate dependencies beyond the receptive field of the GNN ($k=2$ for both methods).
>
> [1] Ketata, et al. "Joint Relational Database Generation via Graph-Conditional Diffusion Models." (2025).
>
> [2] Scassola,et al. "Graph Conditional Flow Matching for Relational Data Generation." (2025).

---

> ### Author Response · Authors · 2025-11-27
>
> **W2) The reviewer points out the influence of preprocessing many-to-many tables on graph generation.** We thank the reviewer for this observation. Note that from the perspective of the relational entity graph, the preprocessed and original representations are informationally equivalent; no information is lost in the transformation. However, we acknowledge that preprocessing two-parent tables influences the inferred partition by altering the node and edge counts used to minimize the description length of the hierarchical SBM. While this effectively modifies the SBM’s optimization objective, our empirical observations indicate no significant drop in the quality of the generated databases w.r.t to the evaluated metrics.  On the other hand, this choice can significantly speed up graph generation (from a few minutes to a few seconds) as it decreases both the node count ($\mathcal{O}(ln^2 |\mathcal{V}|)$) and edge count ($\mathcal{O}(|\mathcal{E}|)$). Therefore, we have revised the text to make this preprocessing step an optional hyperparameter allowing users to improve computational efficiency.
>
> > Q1: How sensitive are results to the neighborhood radius and the subgraph neighbor-sampling scheme during training and inference?
>
> **Q1)** To answer the question, we have extended our ablation study with two additional experiments. The results are robust to both parameters. Regarding neighbor sampling, our ablation shows that limiting the neighborhood size to $S=64$ retains performance (worst-case drops <3%) and, in complex schemas like Berka, can even slightly improve long-range dependency preservation.
>
> Regarding the neighborhood radius, reducing the receptive field to a single hop ($K=1$) yields slightly lower fidelity than our default $K=2$, yet it remains highly competitive. This effectiveness can be explained by the multi-step nature of the diffusion process: information propagates iteratively, effectively expanding the receptive field to $T \times K$ hops over $T$ denoising steps. Consequently, $K=1$ successfully captures long-range dependencies while offering significant computational gains, with over 40% reduction in training and generation time on the Berka dataset. We have added these results and accompanying discussion to our ablation study in Appendix D.4.
>
> | Dataset    | Metric             | RelDiff K=2 | RelDiff S=64         | RelDiff K=1          |
> |-----------|-------------------|---------------------|---------------------|---------------------|
> | **Airbnb** | Shape ($\uparrow$) | 98.14 $\pm$ 0.07     | 98.40 $\pm$ 0.08     | 98.02 $\pm$ 0.04     |
> |            | Trend ($\uparrow$) | 95.76 $\pm$ 0.25     | **96.46 $\pm$ 0.20** | 95.12 $\pm$ 0.31     |
> |            | 1-HOP ($\uparrow$) | **89.37 $\pm$ 0.38** | 88.66 $\pm$ 0.33     | 86.97 $\pm$ 0.37     |
> | **Berka**  | Shape ($\uparrow$) | 97.92 $\pm$ 0.01     | 97.94 $\pm$ 0.02     | **97.83 $\pm$ 0.09** |
> |            | Trend ($\uparrow$) | 98.68 $\pm$ 0.02     | 98.77 $\pm$ 0.04     | 98.91 $\pm$ 0.01     |
> |            | 1-HOP ($\uparrow$) | **96.70 $\pm$ 0.06** | 96.53 $\pm$ 0.05     | 92.48 $\pm$ 0.08     |
> |            | 2-HOP ($\uparrow$) | 95.61 $\pm$ 0.03     | 95.59 $\pm$ 0.04     | **95.59 $\pm$ 0.05** |
> |            | 3-HOP ($\uparrow$) | 89.54 $\pm$ 0.38     | 90.95 $\pm$ 0.15     | 90.41 $\pm$ 0.12     |
> | **F1**     | Shape ($\uparrow$) | **96.80 $\pm$ 0.03** | 96.69 $\pm$ 0.14     | 93.94 $\pm$ 0.05     |
> |            | Trend ($\uparrow$) | 97.35 $\pm$ 0.05     | **97.51 $\pm$ 0.07** | 94.88 $\pm$ 0.02     |
> |            | 1-HOP ($\uparrow$) | **94.02 $\pm$ 0.11** | 93.55 $\pm$ 0.06     | 90.02 $\pm$ 0.64     |
> |            | 2-HOP ($\uparrow$) | **96.19 $\pm$ 0.01** | 96.22 $\pm$ 0.03     | 95.14 $\pm$ 0.42     |
>
>
> > Q2: Many databases encode temporal edges. Can RelDiff model time-stamped relations and reproduce temporal integrity?
>
> **Q2)** We appreciate it if the reviewer could further clarify the definition of temporal integrity? If the reviewer is referring to our model faithfully generating attributed edges, the answer is yes. Attributed edges can be supported by our method in two ways. (i) We can include a pre-processing step to the schema before training the diffusion model, transforming attributed edges into two edges split by a node holding the edge features (in a sense the opposite of W2). (ii) Alternatively, if we utilize a graph neural network that uses edge attributes. we can treat the edge attributes as a “virtual table” performing diffusion updates on the edge attributes as well as the table attributes. Both can be seamlessly integrated into our method. Could the reviewer point us to an example of a publicly available relational database with attributed or timestamped edges? We would be happy to include this as an additional experiment.

---

### Author Response · Authors · 2025-11-27

We thank the reviewers for their insightful feedback and constructive suggestions, which have helped us significantly improve the quality and clarity of our manuscript.

We are encouraged that the reviewers found our modeling choices well-motivated (ghnQ), recognized our thorough experimental evaluation (6VFj), and highlighted the novelty of our D2K+SBM graph generator for preserving relational integrity (5xxZ). The primary concerns centered on the distinctiveness of our approach compared to standard tabular diffusion, the theoretical justification for our generative factorization, and training scalability.

To address these concerns, we have revised the manuscript and conducted additional experiments. Specifically:
- **Novelty**: We clarified the fundamental distinction between RelDiff’s joint graph-based diffusion and standard i.i.d. tabular models. We added a quantitative comparison demonstrating that state-of-the-art tabular diffusion methods fail to capture relational structure (cross-table correlations), whereas RelDiff preserves it effectively (see response to Reviewer 5xxZ). Furthermore, we included a comparison against two concurrent graph-based methods (GRDM, GCFM), showing that RelDiff achieves the strongest performance on 8 out of 11 metrics (see response to Reviewer ghnQ).
- **Scalability & Efficiency**: We optimized our training routine, achieving a 5x speedup (e.g., reducing California training from 17.4h to 3.3h). This effectively matches baselines like ClavaDDPM while retaining our 20x faster sampling speed. We also extended our ablation studies to demonstrate that users can further trade off minimal fidelity for substantial efficiency gains (up to 40% speedup) via subgraph sampling.
- **Theoretical Justification**: We expanded the appendix to justify our choice of the factorization ($p(\mathcal{E})p(\mathcal{V}|\mathcal{E})$), detailing why alternative formulations suffer from intractable combinatorial explosions ($O(N^2)$) unsuitable for real-world databases.

Based on this feedback and our new results, we believe the paper is now much stronger and the method more usable. We have addressed all specific reviewer questions in the detailed responses below.

---

**Summary of Changes** (marked in purple)
- **Optimized Training**: Implemented mixed-precision training and code optimizations, reducing training times by up to 5x (updated Table 10) and added Table 12 reporting memory usage (max 45GB).
- **New Ablation Studies**: Added experiments on subgraph sampling parameters ($S$ and $K$), demonstrating robust performance with reduced neighborhoods offering further speedups ($K=1$ yields ~40% speedup on Berka).
- **Theoretical Clarifications**: Added a new section (Appendix F) analyzing alternative formulations to justify our joint distribution decomposition.
- **Qualitative Evaluation**: Added a new Appendix D.2 with visual comparisons (KDE plots, correlation matrices) of real vs. synthetic data distributions.
- **Schema Flexibility**: Added Figure 9 (AdventureWorks schema) to Appendix E.3 to illustrate the complexity of schemas RelDiff can handle compared to prior work.
- **Preprocessing Flexibility**: Revised the methodology to make the "two-parent" preprocessing step an optional hyperparameter.

---

### Meta-Review · Area_Chair_N61j · 2026-01-06

**Summary:**

The paper proposes RelDiff,  to generate graph data via diffusion. RelDiff combines a joint graph-conditioned diffusion process and a SBM graph generator, i.e. RelDiff consist of resample a D2K+ SBM graph to preserve graph property then apply denoising procedure using heterogeneous GNN. The numerical experiment showed improved performances including benchmark k-hop correlations across 11 real-world databases.

**Reviewer Concerns:**

Novelty is limited. Authors provides additional text to argue the novelty though the concern is only partially addressed.

Comparison with Joint modelling GRDM. Additional experiments are provided.

SBM and the nested version may not express the complex functionals. The concern does not seem to be addressed explicitly.

Temporal edges in the dataset. Additional text clarification provided on the integrating of temporal consideration.

**Reviewer Scores:**

may note change

---

### Decision · Program_Chairs · 2026-01-26

Reject